# BIRD: Behavior Induction via Representation-structure Distillation

**Galen Pogoncheff**
Department of Computer Science
University of California, Santa Barbara
`galenpogoncheff@ucsb.edu`

**Michael Beyeler**
Department of Computer Science, Department of Psychological and Brain Sciences
University of California, Santa Barbara
`mbeyeler@ucsb.edu`

## Abstract

Human-aligned deep learning models exhibit behaviors consistent with human values, such as robustness, safety, and fairness. Transferring these behavioral properties to models trained on different tasks or data distributions remains challenging: aligned behavior is easily forgotten during fine-tuning, and collecting task-specific data that preserves this behavior can be prohibitively costly. We introduce BIRD, a flexible framework for transferring aligned behavior by matching the internal representation structure of a student model to that of a teacher. Applied to out-of-distribution robustness in image classification, BIRD outperforms fine-tuning, transfer learning, and continual learning methods, improving robust accuracy by up to 18% over the next strongest baseline. It remains effective even when the teacher is trained on a much simpler dataset and is $25\times$ smaller in parameter count than the student. In a large-scale study of over 400 teacher-student pairs, we show that three interpretable and computable properties of the teacher's representations explain up to 85% of the variance in transfer success, offering practical guidance for teacher selection and design. We further show that BIRD generalizes beyond applications in vision by enhancing safety alignment in language models when paired with Direct Preference Optimization and improving weak-to-strong generalization when combined with soft-label distillation. BIRD turns small, well-aligned models into scalable alignment seeds, mitigating challenges from key bottlenecks in deploying safe AI systems.

## 1 Introduction

As AI systems become increasingly capable, their alignment with human values (expressed through traits like robustness, safety, and fairness) has become a central challenge (Ortega et al., 2018; Ji et al., 2023). Behavioral alignment typically requires costly supervision: adversarial training, human feedback, or special-purpose datasets (Madry et al., 2017; Ouyang et al., 2022; Rafailov et al., 2023). These techniques do not easily scale to new tasks or domains.

A natural goal is to transfer aligned behavior from one model to another. Yet behavior often degrades during fine-tuning (Shafahi et al., 2019; Qi et al., 2023; Peng et al., 2024), and most transfer methods assume the teacher and student share training data or a common output space (Burns et al., 2023; Zhou et al., 2025). Worse, datasets used to train aligned models are often private.

Recent work in weak-to-strong generalization offers a promising alternative: train a small, well-aligned "weak" model and use it to supervise a larger, more general "strong" model (Burns et al., 2023). However, these approaches still assume the teacher and student share a task, training dataset, or output domain. In this work, we ask: *Can aligned behavior be transferred even when the teacher and student differ in architecture, task, and training data?*

Here, we introduce Behavior Induction via Representation-structure Distillation (BIRD), a simple, drop-in framework for transferring aligned behavior between heterogeneous models by distilling task- and behaviorally-relevant structure from the teacher's representation space into that of the student. BIRD requires no access to teacher training data and succeeds even when the teacher is trained on simpler tasks or domains, enabling scalable reuse of aligned models.

BIRD is inspired by recent work in NeuroAI, where researchers hypothesize that desirable behavioral properties (e.g., robustness to noise, transformation invariances) may be encoded in the geometry of brain representations (Chung & Abbott, 2021; Zador et al., 2023; Mineault et al., 2024). Several studies have operationalized this idea by explicitly aligning deep network representations with neural recordings, showing that representation-level alignment can improve robustness and generalization on image classification benchmarks (Dapello et al., 2023; Safarani et al., 2021; Li et al., 2019). Related work has also examined how representational similarity between modern vision models and the biological visual system can be quantified across differing objectives and architectures, providing tools for comparing internal geometries even when tasks diverge (Pogoncheff et al., 2024). However, these approaches typically require access to neural recordings and often rely on shared stimulus domains. Moreover, gains achieved through neural alignment can degrade when transferring to categories or domains outside the alignment dataset (Dapello et al., 2023).

Our work builds on the core insight that behavioral properties are encoded in the structure of a model's latent representations (Zou et al., 2023; Schneider et al., 2025), but generalizes it in two key ways. First, BIRD does not rely on biological recordings or shared datasets; the teacher and student may differ in size, architecture, domain, and output space. Second, we empirically identify three computable properties of a teacher's representation space that reliably predict transfer success. This enables principled selection of both teacher models and representation layers.

We first evaluate BIRD in the context of out-of-distribution robustness transfer in image models. Studying over 400 teacher-student pairs varying in architecture, capacity, and training data, we find:

- BIRD outperforms transfer methods including fine-tuning, continual learning, and activation-based distillation, improving robust accuracy by up to 18% over the strongest baseline.

- BIRD enables weak-to-strong transfer from small, simple teachers (e.g., CIFAR-10-trained MobileNetV2) to students up to $25\times$ larger, trained on more complex datasets (e.g., TinyImageNet).

- Transfer success is predictable and actionable: three interpretable properties of the teacher's representation space that quantify task and behavioral relevance explain up to 85% of the variance in transfer outcomes, offering practical guidance for alignment layer and teacher selection.

We further show that BIRD extends beyond vision. In language models, supplementing Direct Preference Optimization (DPO) with BIRD improves the efficacy of safety alignment on the PKU-SafeRLHF dataset, and enhances weak-to-strong generalization when combined with soft-label distillation.

By aligning representation structure instead of activation values or model outputs, BIRD provides a flexible mechanism for transferring aligned behavior across models and domains. This design makes BIRD more general than existing approaches in knowledge distillation (Hinton et al., 2015; Qiu et al., 2022; Jin et al., 2024) and continual learning (Li & Hoiem, 2017; Shafahi et al., 2019) that assume access to teacher data or rely on shared domains, output spaces, or tasks between teacher and student. It advances the promise of weak-to-strong generalization and sets the stage for scalable alignment across tasks.

## 2 RELATED WORK

### 2.1 SCALABLE ALIGNMENT VIA WEAK-TO-STRONG SUPERVISION

Behavioral misalignment arises when models optimize their training objective while diverging from human intent (Amodei et al., 2016; Di Langosco et al., 2022; Razin et al., 2024). Current approaches to mitigate this, such as preference tuning for language models (Ouyang et al., 2022; Rafailov et al., 2023) or adversarial training for vision models (Madry et al., 2017; Hendrycks et al., 2021), are effective work but require costly, task-specific datasets and supervision, limiting scalability.

*Scalable oversight* aims to reduce this dependence by developing methods for supervising advanced models efficiently, even when human feedback is limited or unavailable (see Ji et al., 2023). A promising direction is weak-to-strong generalization, in which a small, well-aligned model provides soft-label supervision to guide the training of a larger, more capable model (Burns et al., 2023; Zhu et al., 2024; Zhou et al., 2025). Our work generalizes this paradigm. Instead of requiring shared output spaces and access to a teacher's training data (even if unlabeled), we transfer behavior between heterogeneous models by leveraging the representation structure of a teacher model as a supervisory signal. This enables weak-to-strong transfer across differing tasks, datasets, and architectures.

## 2.2 LIMITS OF ROBUSTNESS TRANSFER IN IMAGE MODELS

Adversarial robustness is a primary benchmark for studying behavioral transfer (Shafahi et al., 2019; Nern et al., 2023; Liu et al., 2023; Xu et al., 2023). While classical approaches require expensive, online generation of adversarial examples during training (Madry et al., 2017; Schmidt et al., 2018), recent work has explored *transferring* robustness to new tasks without retraining from scratch.

A key challenge is *catastrophic forgetting* (Kirkpatrick et al., 2017). When robust models are fine-tuned on new data, robustness is often lost (Shafahi et al., 2019; Nern et al., 2023). To address this, methods have incorporated adversarial examples during transfer or added constraints to limit feature drift (Chen et al., 2021; Fan et al., 2021; Liu et al., 2023; Xu et al., 2023). Shafahi et al. (Shafahi et al., 2019) proposed adversarially robust transfer learning using Learning without Forgetting (LwF) (Li & Hoiem, 2017), which preserves robustness by constraining changes to the final-layer features. However, these techniques assume that robust features generalize across domains, which holds only with large and diverse pretraining datasets. In contrast, our work enables behavior transfer from smaller, simpler models trained on low-resource domains, without requiring shared inputs or labels.

## 2.3 DISTILLING REPRESENTATION GEOMETRY

Knowledge distillation (KD) enables a student model to learn from a teacher by mimicking outputs or hidden activations. Originally developed for model compression (Bucilua et al., 2006; Hinton et al., 2015), available KD methods have since expanded to include intermediate-layer supervision (Romero et al., 2014; Zagoruyko & Komodakis, 2016) and cross-modal transfer (Gupta et al., 2016). These methods assume shared tasks and output spaces and often transfer sample-level information.

BIRD takes a different approach. Rather than matching outputs or activations, we transfer the pairwise structure of the teacher's representation space, captured via Gram matrices over input batches. This quantifies the geometry of internal representations, rather than specific activation values. This builds on the idea that a model's knowledge lies not only in its outputs, but in the organization of its representation space (Hjelm et al., 2018; Tian et al., 2019; Muttenthaler et al., 2024). While prior work uses this idea for unsupervised learning or intra-task alignment (Qiu et al., 2022), BIRD applies it to cross-task, cross-domain behavior transfer, generalizing KD into a scalable mechanism for aligned supervision.

## 2.4 REPRESENTATION ALIGNMENT IN NEUROAI

Work in NeuroAI suggests that robust and general behavior in biological systems may arise from the structure of neural representations (Chung & Abbott, 2021; Zador et al., 2023). Several studies have attempted to bias AI models toward brain-like representations by minimizing dissimilarity to neural recordings from visual cortex, often using objectives based on Centered Kernel Alignment (CKA) or Representational Similarity Analysis (RSA) (Dapello et al., 2023; Safarani et al., 2021; Muttenthaler et al., 2024). These approaches have shown moderate improvements in robustness but typically require neural data, assume shared input domains, and show limited generalization.

Our work draws inspiration from these ideas but removes the need for brain recordings or stimulus overlap. BIRD operationalizes the hypothesis that structured representations support general behavior, using any aligned model as a teacher. By enabling behavior transfer across architectures and domains, BIRD extends NeuroAI insights into a general-purpose framework for scalable alignment.

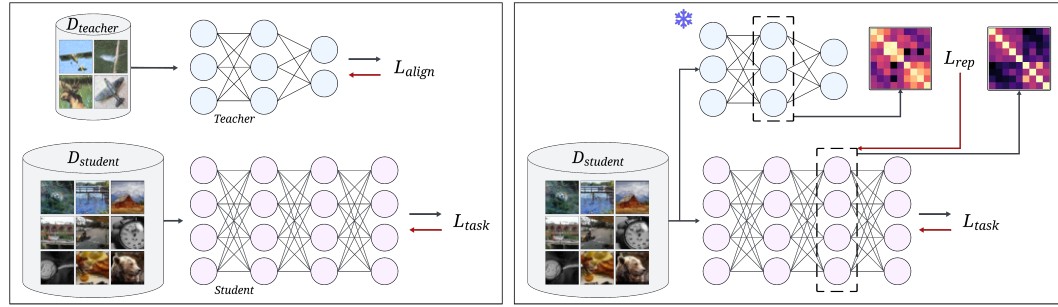

(a) Independent model pre-training     (b) Layer selection + representation-structure distillation

Figure 1: Overview of BIRD: **(a)** First, a student model is pre-trained on its target training set, $\mathcal{D}_{\text{student}}$. A teacher model is independently trained to develop aligned behavior on $\mathcal{D}_{\text{teacher}}$ (optimizing $\mathcal{L}_{\text{align}}$). **(b)** Next, the student model is fine-tuned on its original dataset to maintain task performance while learning latent representations with similar structure to that of the (frozen) teacher.

## 3 BIRD: BEHAVIOR INDUCTION VIA REPRESENTATION-STRUCTURE DISTILLATION

We introduce BIRD, a flexible framework for transferring aligned behavior from a teacher model to a student model. Our approach is grounded in the hypothesis that task-general behavioral properties (i.e., robustness, safety, and invariance) are encoded in the structure of a model's internal representation space, as suggested in Zou et al. (2023). We posit that guiding a student model to adopt similar representational structure to its teacher biases it toward learning the same aligned behaviors.

BIRD proceeds in three steps (Figure 1):

1. We assume access to a trained teacher model $g_\phi : \mathcal{D}_{\text{teacher}} \to \mathcal{Y}_{\text{teacher}}$ that exhibits desirable behavioral properties, and a pretrained student model $f_\theta : \mathcal{D}_{\text{student}} \to \mathcal{Y}_{\text{student}}$ for which we wish to induce those properties (Figure 1a).
2. A guiding layer in the teacher and a guided layer in the student are selected for distillation based on their relevance to task and behavioral alignment (Section 6).
3. The student is fine-tuned to preserve performance on its original task while learning a representation space whose structure mimics that of the teacher (Figure 1b).

To implement this, we define a loss that combines task performance and representational alignment:

$$\mathbb{E}_{B \sim \mathcal{D}_{\text{student}}} \left[ \alpha \mathcal{L}_{\text{task}} \big( f_\theta(B), \cdot \big) + \beta \mathcal{L}_{\text{rep}} \big( u(B), v(B) \big) \right] \tag{1}$$

Here, $B$ is a batch of inputs from the student's training distribution while $\alpha$ and $\beta$ are hyperparameters that weight the relative contributions of task and representation-structure loss. The functions $u$ and $v$ map those inputs to intermediate layer representations in the teacher and student, respectively. The first term, $\mathcal{L}_{\text{task}}$, is the task-specific loss that the student was originally trained to minimize (e.g., cross-entropy). The second term, $\mathcal{L}_{\text{rep}}$, penalizes dissimilarity in representation structure:

$$\mathcal{L}_{rep}(u(B), v(B)) = 1 - \text{CKA}_{\text{linear}}(u(B), v(B)). \tag{2}$$

We use CKA (Kornblith et al., 2019) to quantify the alignment of pairwise similarity structures within a batch of inputs:

$$\text{CKA}_{\text{linear}}(u(B), v(B)) = \frac{||v(B)^T u(B)||_{\text{F}}^2}{||u(B)^T u(B)||_{\text{F}}^2 \cdot ||v(B)^T v(B)||_{\text{F}}^2}. \tag{3}$$

We select CKA as a measure of representational similarity given its (i) proven effectiveness in comparing deep network representations, (ii) reliability when comparing high-dimensional representation spaces, and (iii) ease of interpretation. Compared to common losses used in KD, such as $L_2$ or KL-divergence, this CKA-based objective aligns the geometry of teacher and student representation manifolds rather than enforcing instance-level similarities. Because CKA evaluates pairwise similarities across a batch (as opposed to per-example similarities, which are closely tied to the teacher's

Table 1: Accuracy (%) of MobileNetV2 (MN2), ResNet18 (RN18), DenseNet169 (DN169), and Vision Transformer (ViT) models after behavior transfer using clean data from CIFAR-10 (C10), CIFAR-100 (C100), TinyImageNet (TIN), and ImageNet (IN). Values reported are accuracy over all clean and corrupted images from the target test set, averaged over 3 seeds.

| Model | Source Data | Target Data | Accuracy of Behavior Transfer Method (↑) | | | | | | |
|---|---|---|---|---|---|---|---|---|---|
| | | | None | LP | FT | LP-FT | Hints | LwF | BIRD |
| MN2 | C10 | C100 | 51.31 | 10.95 | 51.12 | 47.56 | 51.53 | 52.21 | **54.77** |
| | C10 | TIN | 20.74 | 5.24 | 20.00 | 18.12 | 21.27 | 20.52 | **24.11** |
| | C100 | TIN | 20.74 | 18.84 | 20.66 | 23.52 | 21.26 | 23.18 | **25.03** |
| | C10 | IN | 22.59 | 1.29 | 22.50 | 22.36 | 22.25 | 23.23 | **23.38** |
| | C100 | IN | 22.59 | 6.18 | 22.36 | 22.34 | 22.49 | 23.27 | **23.68** |
| RN18 | C10 | C100 | 52.03 | 16.93 | 51.95 | 50.63 | 52.20 | 55.42 | **57.39** |
| | C10 | TIN | 20.56 | 7.25 | 20.10 | 19.43 | 20.92 | 22.17 | **23.60** |
| | C100 | TIN | 20.56 | 20.95 | 20.75 | 23.66 | 20.71 | 24.48 | **24.49** |
| | C10 | IN | 20.96 | 5.48 | 21.02 | 20.62 | 22.66 | 22.25 | **23.16** |
| | C100 | IN | 20.96 | 9.93 | 20.96 | 20.42 | 22.44 | 22.25 | **23.52** |
| DN169 | C10 | C100 | 54.51 | 23.92 | 55.84 | 53.39 | 54.92 | 56.92 | **59.04** |
| | C10 | TIN | 22.59 | 10.66 | 23.39 | 21.20 | 22.68 | 24.14 | **25.25** |
| | C100 | TIN | 22.59 | 23.55 | 23.19 | 24.86 | 22.75 | 26.14 | **27.46** |
| | C10 | IN | 26.46 | 2.39 | 26.86 | 26.70 | 26.59 | 27.08 | **27.43** |
| | C100 | IN | 26.46 | 7.25 | 26.18 | 26.16 | 26.78 | 27.21 | **27.76** |
| ViT | C10 | C100 | 50.77 | 40.07 | 53.83 | **54.09** | 53.57 | 51.32 | 53.71 |
| | C10 | TIN | 22.08 | 20.28 | 25.82 | 25.56 | 25.53 | 22.28 | **25.98** |
| | C100 | TIN | 22.08 | 22.94 | 24.24 | 23.91 | 25.82 | 22.36 | **27.26** |
| | C10 | IN | 25.83 | 8.71 | 25.36 | 25.10 | 26.20 | 26.15 | **26.32** |
| | C100 | IN | 25.83 | 10.18 | 24.78 | 24.68 | 26.15 | 26.01 | **26.28** |

specific outputs), it captures higher-order relationships that reflect general, behaviorally relevant properties of the teacher's representations (Kornblith et al., 2019).

Our approach draws inspiration from recent work in neuroscience-informed representation learning, where deep networks are trained to jointly minimize a task loss and a neural alignment loss based on brain recordings (Li et al., 2019; Federer et al., 2020; Safarani et al., 2021; Dapello et al., 2023). Unlike those approaches, BIRD does not require neural data or shared stimulus distributions.

A central distinction between BIRD and prior approaches to distillation and behavior transfer lies in its flexibility across heterogeneous models and tasks. Teacher supervision in BIRD comes solely from representation structure, measured via CKA. This design enables BIRD to support behavior transfer in settings where direct supervision or shared objectives are unavailable. Concretely:

- **BIRD does not require a shared input domain, output space, or task** between teacher and student models.
- **BIRD does not rely on paired inputs or access to the teacher's training data**; supervision is provided by projecting the student's inputs through the teacher's representation space.

Together, these properties set BIRD apart as a practical and general framework for transferring aligned behavior beyond the constraints of existing approaches in KD.

## 4 TRANSFERRING ROBUSTNESS ACROSS DATASETS AND ARCHITECTURES

We first evaluate BIRD in the context of robust image classification, where the goal is to transfer out-of-distribution (OOD) robustness from a teacher model trained on a lower-complexity dataset $\mathcal{D}_{\text{teacher}}$ to a student model trained on clean data from a higher-complexity dataset $\mathcal{D}_{\text{student}}$ ($\mathcal{D}_{\text{student}} \neq \mathcal{D}_{\text{teacher}}$). This setting follows the aims of weak-to-strong generalization, where aligned behavior is induced in a larger, less specialized model using supervision from a smaller, aligned one.

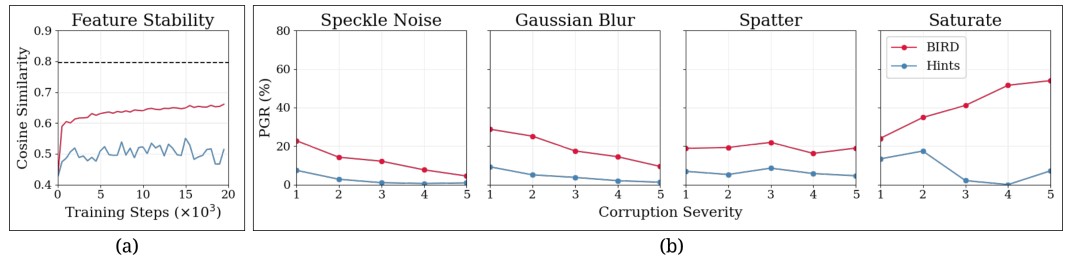

Figure 2: Comparing transferred robustness between BIRD and Hints students (CIFAR-10 trained teacher, TinyImageNet student). **(a)** Feature stability (average cosine similarity between clean images and their corrupted versions) measured over first 20,000 steps of training. Feature stability of teacher shown as horizontal dashed line. **(b)** PGR measured over each corruption type.

**Setup and evaluation protocol**   We consider robust classification transfer across five dataset pairs: CIFAR-10 → CIFAR-100, CIFAR-10 → TinyImageNet, CIFAR-100 → TinyImageNet, CIFAR-10 → ImageNet, and CIFAR-100 → ImageNet (Krizhevsky et al., 2009; Deng et al., 2009). Teachers are trained to be robust to 15 ImageNet-C corruptions (Hendrycks & Dietterich, 2019), while students see only clean images. To maintain consistency across all dataset pairs, all images are resized to $32 \times 32$ pixels.

Robustness is measured using accuracy over the clean and corrupted test set and Performance Gap Recovered (PGR) (Burns et al., 2023):

$$\text{PGR} = \min \left( \max \left( 0, \frac{\text{Acc}_{\text{post}} - \text{Acc}_{\text{pre}}}{\text{Acc}_{\text{ceiling}} - \text{Acc}_{\text{pre}}} \right), 1 \right), \tag{4}$$

where $\text{Acc}_{\text{post}}$ is student accuracy after behavior transfer, $\text{Acc}_{\text{pre}}$ is the pre-transfer baseline, and $\text{Acc}_{\text{ceiling}}$ is the accuracy of a student trained directly with access to OOD corruptions.

We test BIRD across four architectures: MobileNetV2 (Sandler et al., 2018), ResNet18 (He et al., 2016), DenseNet169 (Huang et al., 2017), and Vision Transformers (Dosovitskiy et al., 2020). We compare BIRD to five baseline strategies that do not access corrupted target training data: linear probing (LP), full fine-tuning (FT), sequential LP followed by FT (LP-FT) (Kumar et al., 2022; Nern et al., 2023), LwF (Li & Hoiem, 2017; Shafahi et al., 2019), and hint-based distillation (Hints) (Romero et al., 2014), which aligns activation values between teacher and student via linear mapping and $L_2$-norm representation loss. Additional training details, corruption visualizations, and evaluation breakdowns are provided in Appendix A.1.

**Comparison with baselines**   Across nearly all dataset pairs and model architectures, BIRD achieves the highest out-of-distribution robustness and PGR (Table 1). For instance, when transferring robustness from CIFAR-10 to CIFAR-100, BIRD, on average, improves robustness by 4.5 percentage points and recovers 31.8% of the performance gap to the robustness ceiling. The next best method, LwF, recovers only 13.5%. Similar trends are observed in the CIFAR-100 → TinyImageNet (25.2% vs. 13.8%), CIFAR-10 → TinyImageNet (22.4% vs. 4.9%), CIFAR-10 → ImageNet (8.7% vs. 4.2%), and CIFAR-100 → ImageNet (10.5% vs. 3.7%) transfers.

LP and FT alone fail to consistently improve robustness over the baseline. LP-FT provides modest gains, but only in a select few architecture, dataset pairs. This highlights a key challenge of behavior transfer from weak sources: robust but highly specific features may not generalize across distribution shifts and are therefore easily forgotten when no structural constraint are imposed during transfer.

**Comparison with activation matching**   To isolate the effect of distilling representational structure, we compare BIRD to Hints, which supervises the student via linear mapping and $L_2$-norm on the same representation layers. Despite identical teacher-student pairs and alignment points, BIRD achieves higher robustness in every setting and corruption category (Figure 2). This suggests that representation structure (captured via CKA) encodes more generalizable behavioral information than raw, sample-level activation values. We conclude that BIRD's success stems not just from where it supervises, but from how it supervises.

**Scaling to larger student models**   We next test whether BIRD generalizes to student models of higher capacity than the teacher. Fixing a MobileNetV2 teacher, we apply BIRD to a series of student architectures of varying capacity. As shown in Figure 3, robustness improves across all model sizes, including a 22.4% PGR for the ResNet152 student, despite it having $25\times$ more parameters than the teacher. These results confirm BIRD's effectiveness for weak-to-strong behavior transfer, even in extreme capacity mismatches. As is common in robust learning, improvements in OOD robustness are sometimes accompanied by minor reductions in clean accuracy. Detailed clean-vs-corruption breakdowns and per-seed results are provided in Appendix A.1.

## 5   ABLATION STUDIES

**Effect of batch size on behavioral transfer**   In BIRD, representation structure is quantified over batches of training data. Although CKA is capable of providing meaningful estimates of representational similarity even when the number of samples is smaller than the dimensionality of the representations (Kornblith et al., 2019), undersampling a representation space with too small of a batch size may fail to capture nuances of its structure, especially as it relates to model behavior. Appendix Figure 6 shows the effects of batch size on behavioral transfer performance in ResNet18 models in three transfer settings (transferring robust behavior from CIFAR-10 → CIFAR-100, CIFAR-10 → TinyImageNet, and CIFAR-100 → TinyImageNet). We find that while smaller batch sizes (e.g., 32 and 64) still yield performance improvements in robust classification over the original student model, these gains were far less than could be achieved with larger batch sizes, as expected. In Sections 4 and 6, we maintained a consistent batch size of 128 across all evaluated methods. However, the results of this ablation study suggest that increasing the batch size for BIRD could yield even stronger behavioral transfer performance.

**Choice of CKA kernel function**   Throughout this work, we use CKA with a linear kernel to quantify representational similarity, given its extensive use in analyzing deep learning representations (Kornblith et al., 2019; Nguyen et al., 2020; Raghu et al., 2021) and its ease of interpretation. Although we focus on the linear variant, CKA naturally extends to alternative kernels. In follow-up experiments, we observe that using an RBF kernel yields comparable improvements in robust behavior transfer (Appendix Table 4). More broadly, the key contribution of BIRD is a flexible distillation framework grounded in representation structure. In settings where specific geometric properties of the representation space are known to be behaviorally relevant a priori, practitioners may benefit from selecting kernels accordingly.

## 6   WHAT MAKES A GOOD TEACHER FOR BEHAVIOR TRANSFER?

The preceding experiments demonstrated BIRD's ability to transfer robust behavior across datasets and architectures. Here, we investigate the conditions under which this transfer is most successful. Specifically, we want to understand what makes a teacher a good source for behavior transfer.

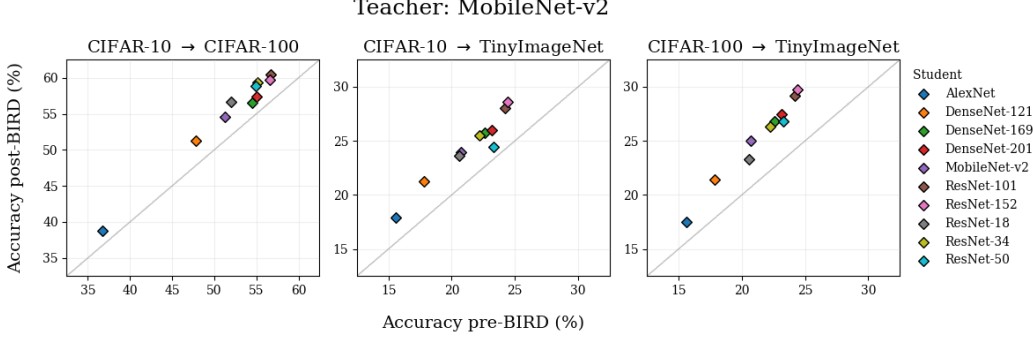

Figure 3: OOD robustness of student models before and after BIRD, guided by a MobileNet teacher trained on a lower-complexity dataset. BIRD improves robustness across all student capacities.

**Setup and analysis framework** Our analysis is based on the hypothesis that transfer success depends on two measurable properties of the teacher's representation space: *Task relevance*, the degree to which the teacher's representations are informative for the student's downstream task, and *Behavioral relevance*, the extent to which the teacher's representations support aligned behavior.

We construct a pool of 144 teacher models varying along these two axes. Teachers are trained on one of three datasets (CIFAR-10, CIFAR-100, or TinyImageNet) to manipulate task relevance. Behavioral relevance is varied by augmenting training data with randomly selected subsets of ImageNet-C corruptions. Teachers span four architectures: AlexNet (Krizhevsky et al., 2012), ResNet18 (He et al., 2016), DenseNet121 (Huang et al., 2017), and MobileNetV2 (Sandler et al., 2018).

Each teacher is used to supervise a ResNet50 student trained on one of three datasets (CIFAR-10, CIFAR-100, or TinyImageNet), resulting in 432 teacher-student pairs. We quantify properties of each teacher's representation space using established metrics (Alain & Bengio, 2016; Ilyas et al., 2019) that assess downstream utility of a feature space:

- **Task relevance** We train linear probes on the teacher's representation using clean data from $\mathcal{D}_{student}$ and measure (i) *Probing accuracy:* classification accuracy of the linear probe (Alain & Bengio, 2016) on held-out student data (ii) *Complementary knowledge:* fraction of student samples correctly classified by the teacher's probe but not by a probe trained on the student's own representation (Roth et al., 2023).
- **Behavioral relevance** We aggregate the $\gamma$-robust usefulness of each feature (Ilyas et al., 2019), which measures whether features retain predictive value under corruptions. Full computation details are provided in Appendix A.4.

**Explaining transfer success** For each student dataset, we fit a linear model to predict accuracy on the corrupted test set using the three metrics described above. We report $R^2$ values to quantify explained variance in behavior transfer from these properties of the teacher's representation space.

**Results** Impressively, these simple, interpretable properties explain the vast majority of transferred robustness. We observe strong predictive power across all student datasets. For students trained on TinyImageNet, the model explains 81.8% of the variance in PGR; for CIFAR-100, 85.5%; and for CIFAR-10, 73.6% (Appendix, Figure 7). Across all student datasets, the most predictive single factor is the behavioral relevance of the teacher's representation space, as measured by aggregated $\gamma$-robust usefulness, explaining more than 50% of the variance in PGR alone. Task relevance metrics (probing accuracy and complementary knowledge) add additional explanatory power, especially when transferring to more complex student datasets like TinyImageNet.

**Implications for teacher selection** These findings provide actionable guidance for selecting or training effective teacher models. When choosing among candidate teachers, we recommend prioritizing those with high behavioral relevance (i.e., the behavior of interest is strongly encoded in the teacher's representation space), even if they were trained on a different dataset or have limited task overlap with the student.

## 7 TRANSFERRING BEHAVIOR IN LANGUAGE MODELS

To assess the generality of BIRD beyond robustness transfer in vision, we evaluate its application in two distinct language modeling settings: safety alignment and weak-to-strong generalization.

### 7.1 TRANSFERRING SAFE BEHAVIOR

*Safety alignment*, fine-tuning models so that their outputs are safe with respect to human-defined criteria, is a central challenge in alignment research. A common approach is DPO (Rafailov et al., 2023), which optimizes alignment using paired preferred and rejected responses to the same query.

We investigate whether supplementing DPO with BIRD can further improve safety alignment. To do this, we encourage a generative student model to adopt a representational structure similar to that of a simple teacher classifier trained to distinguish safe from unsafe responses (Appendix A.7.2).

**Setup and Evaluation Protocol**    We evaluate safety alignment on the 135M and 360M parameter variants of SmolLM2, state-of-the-art, "small" language models that are accessible within our compute budget (Allal et al., 2025). For preference supervision, we sample $10,000$ examples from the PKU-SafeRLHF dataset (Ji et al., 2024a), which contains prompts paired with safe and unsafe responses, as evaluated across 19 safety dimensions (Appendix A.7.1). Student models are trained with two strategies: (i) *DPO* and (ii) DPO supplemented with BIRD loss (*DPO+BIRD*). In the latter setting, student representation structure is supervised by that of a simple binary classifier fine-tuned on PKU-SafeRLHF-QA (Ji et al., 2024b) dataset to predict whether or not a given response is safe (risk-neutral according to all 19 safety dimensions). To evaluate safety alignment, we report the percentage of generated responses to queries from the PKU-SafeRLHF test set judged as safe by an independently trained safety classifier (Appendix A.7.3).

**Results**    Experimental results are summarized in Table 2. Across three seeds, all models fine-tuned on the safety preference dataset outperform the baseline instruction tuned model (None), and models trained with BIRD further produce a higher proportion of safe responses than DPO alone. Sample responses are provided in Appendix A.7.4. Although relatively small language models were studied, it demonstrates BIRD's generalization beyond robustness transfer in vision to the challenge of safety alignment in language models. Future work will extend these analyses to larger models.

Table 2: Safety alignment performance of generative models on the PKU-SafeRLHF test set. *% Safe*: percentage of query responses from that model evaluated as safe according to an LLM judge.

| Student | % Safe (↑) | | |
|---|---|---|---|
| | None | DPO | DPO+BIRD |
| SmolLM2-135M-Instruct | 43.88 | 65.48 | **71.28** |
| SmolLM2-360M-Instruct | 47.63 | 86.57 | **88.37** |

## 7.2    EXTENDING SOFT-LABEL WEAK-TO-STRONG GENERALIZATION WITH BIRD

We further assess the generality of BIRD in the context of weak-to-strong generalization (Burns et al., 2023), in which we seek to distill capabilities from a low-capacity, aligned teacher into a higher-capacity, unaligned student model without access to ground-truth labels.

**Setup and evaluation protocol**    We adopt the setup introduced in Burns et al. (2023), in which a small, aligned teacher model is fine-tuned on a target dataset and then used to supervise a larger student. We use GPT2-Small as the teacher and either GPT2-Medium or GPT2-Large as the student.

We consider three multiple-choice question-answering datasets: **SciQ** (Johannes Welbl, 2017): science questions with factual answers; **BoolQ** (Clark et al., 2019): yes/no questions based on paragraph context; **Cosmos QA** (Huang et al., 2019): commonsense inference over short passages.

Each student is trained using two supervision strategies: (i) *Soft-label distillation*: cross-entropy loss between student predictions and the teacher's soft probabilities, (ii) *Soft-label + BIRD*: soft-label loss, accompanied with BIRD's representation-structure loss computed over the final token embedding layer of teacher and student. Following Burns et al. (2023), we report PGR on a held-out test set relative to a ceiling set by fine-tuning of the student on target data with ground-truth labels.

**Results**    Table 3 summarizes performance across all dataset and model combinations. In three of the six configurations, adding BIRD to soft-label distillation yields a noticeable improvement in behavior transfer. On BoolQ, neither soft-label distillation nor BIRD outperform the weak teacher (0% PGR). This suggests that not all aligned behaviors captured by small models generalize to more complex reasoning tasks or that representational distillation at the final layer may not be sufficient in this domain. This highlights a broader opportunity: combining BIRD with additional alignment signals (e.g., multi-layer supervision) may offer more consistent benefits in complex domains.

Table 3: PGR (averaged over 3 seeds) for GPT2-Medium and GPT2-Large models trained on soft-labels from GPT2-Small (Soft-Label) or soft-labels with BIRD (+BIRD).

| Dataset | Student | % PGR (↑) | |
|---|---|---|---|
| | | Soft-Label | +BIRD |
| SciQ | GPT2-Medium | 7.79 | **16.14** |
| | GPT2-Large | 17.70 | **24.19** |
| Cosmos QA | GPT2-Medium | **47.00** | 42.76 |
| | GPT2-Large | 65.51 | **68.02** |

## 8 DISCUSSION

We introduce Behavior Induction via Representation-structure Distillation (BIRD), a simple and general framework for transferring aligned behavior by distilling the structure of a teacher model's internal representations. Unlike prior approaches, BIRD does not require the teacher and student to share common training data or domains, output spaces, or architectures between teacher and student. Our experiments in robust image classification show that BIRD consistently outperforms existing transfer methods, even when the teacher is significantly smaller and trained on a simpler dataset. We further identify three interpretable properties of the teacher's representation space that strongly predict transfer success. Finally, we show that BIRD provides complementary gains when paired with existing methods for safety alignment and weak-to-strong generalization in language models. These results are not intended as new state-of-the-art benchmarks, but rather as evidence of BIRD's generality and its ability to enhance existing approaches without requiring new data or reward modifications.

**Key takeaways** BIRD contributes two key advances. First, it establishes a flexible, drop-in mechanism for aligned behavior transfer that operates across domains, architectures, and label spaces by supervising over representation structure. Second, it offers a systematic analysis of what makes a good teacher for behavior transfer. Our findings suggest that small and simple models can serve as effective alignment scaffolds for larger, unaligned models if their representations are behaviorally relevant. BIRD is especially useful in settings where aligned behavior must be scaled without re-training large models or access to private data.

**Limitations and future directions** Extending BIRD to additional behaviors beyond robustness and safety (e.g., honesty) and to model-pairs that differ in modality is a promising direction that we are excited to investigate in future work. We provide initial evidence for the utility of BIRD when used alongside foundation vision-language encoders in Appendix A.6. Additionally, transfer success is likely bounded by student capacity and task complexity; future work should quantify this saturation effect more precisely (Appendix A.5). Further, we currently supervise alignment at a single layer selected by a heuristic. While our findings suggest that exact layer choice is not overly sensitive (Appendix A.1.4), future work may explore multilayer extensions to capture deeper structural alignment. We expect that the primary challenges that exist with such a strategy include selecting optimal alignment layer pairs and selecting the weight that each layer pair contributes to the BIRD representation loss (challenges that we were able to approach in single-layer alignment by probing properties of candidate alignment layers and balancing task and representation losses (Appendix A.1.4 and A.1.3)). We perform an initial multi-layer study, naively selecting layers at four regularly-spaced model depths and weighing representation loss equally across alignment layers in Appendix A.3. Finally, our behavioral relevance metric is tailored to robust classification; in other domains, tools like linear tomography (Zou et al., 2023) or causal mediation analysis (Vig et al., 2020) may help localize behavior-relevant representations.

## ETHICS STATEMENT

BIRD is intended as a tool to improve human-AI alignment. However, its effectiveness depends on the teacher: if the teacher encodes biased, harmful, or misaligned behaviors, BIRD may transfer

those behaviors to the student. Ensuring the integrity of teacher models is therefore essential for responsible deployment.

## REPRODUCIBILITY STATEMENT

We make a public implementation of BIRD available at `github.com/gpogoncheff/bird`. Training and evaluation configuration details required for reproduction (e.g., choice of optimizers, data augmentations, and hyperparameters) are provided in the appendix.

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

# A APPENDIX

## A.1 EVALUATING ROBUST BEHAVIOR TRANSFER

### A.1.1 OUT-OF-DISTRIBUTION CORRUPTIONS FROM THE IMAGENET-C BENCHMARK

The ImageNet-C benchmark (Hendrycks & Dietterich, 2019) is a standardized suite for evaluating the robustness of image classification models to common corruptions. These image corruptions are algorithmically generated to simulate four different categories of real-world sources of image corruption (noise, blurring, weather effects, and digital effects). These corruptions can be readily applied to images from datasets other than ImageNet (e.g., CIFAR-10, CIFAR-100, and TinyImageNet), as was done in this work. There are 19 total corruption types, each from one of these four corruption categories (Figure 4). Each corruption is additionally applied at five levels of severity, reflecting of the magnitude of the corruption (Figure 5). In all experiments of Section 4, teacher models are trained to be robust to 15 of these corruption types ("gaussian noise", "shot noise", "impulse noise", "defocus blur", "glass blur", "motion blur", "zoom blur", "snow", "frost", "fog", "brightness", "contrast", "brightness", "pixelate", and "jpeg compression") and, after robust transfer, students are evaluated on the remaining 4 four held out corruption types ("speckle noise", "gaussian blur", "spatter", and "saturate"), one from each corruption category, to evaluated generalized robustness (Hendrycks & Dietterich, 2019).

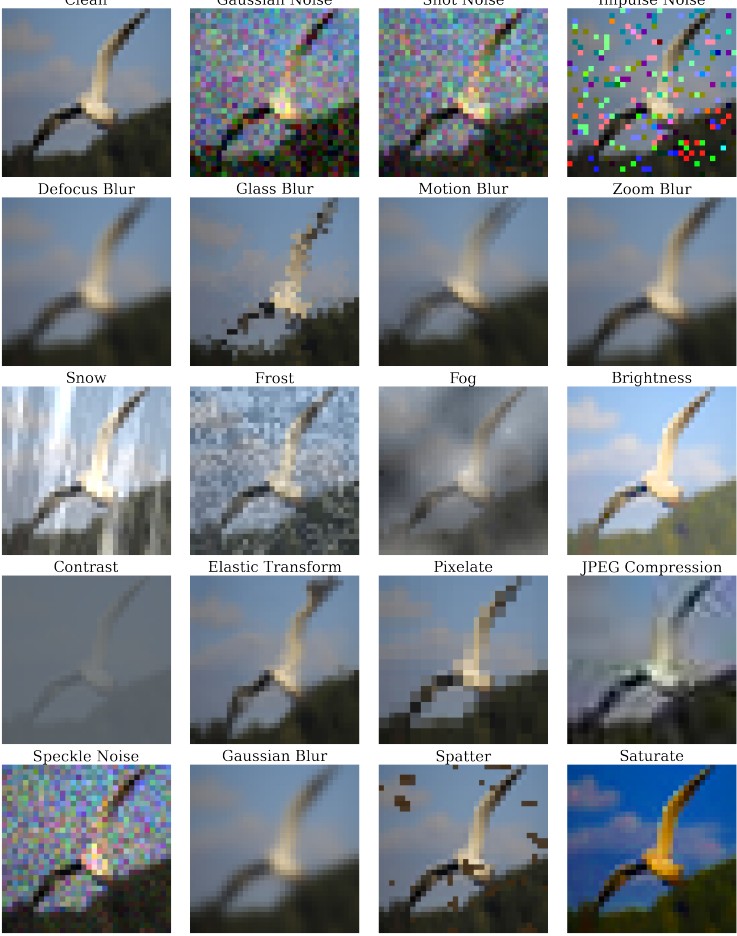

Figure 4: Sample CIFAR-10 image distorted with corruptions from the ImageNet-C benchmark (Hendrycks & Dietterich, 2019). "Clean" designates the original, uncorrupted image. Corruptions are depicted at maximal severity.

Corruption Type: Speckle Noise

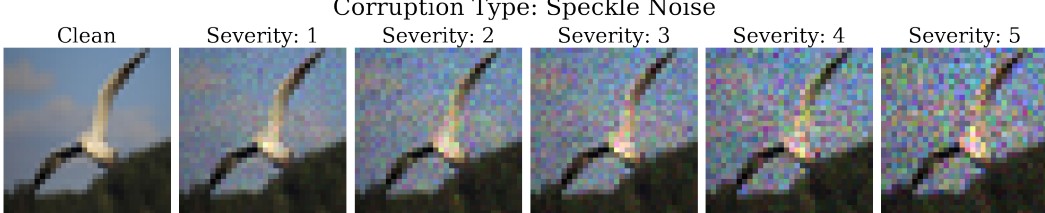

Figure 5: Sample CIFAR-10 image distorted with "Speckle Noise" corruption at severities 1-5.

### A.1.2 ROBUST TRANSFER BASELINES

In Section 4, we seek to compare robust behavior transfer with BIRD to robust transfer learning baselines that (1) solely rely on learning from clean data during the transfer learning task and (2) do not make inherent assumptions about the exact robust pre-training method. These two criteria strongly reflect the behavior transfer paradigm that is studied in this paper: transferring behavior of an aligned model to a new, target domain using only the data that we already have available for the target domain. To the best of our knowledge, the most relevant and highest performing methods that meet these criteria are robust transfer learning using continual learning strategies (LwF) (Shafahi et al., 2019), Linear Probing (LP), Fine-tuning (FT), and Linear Probing followed by Fine-tuning (LP-FT) (Shafahi et al., 2019; Kumar et al., 2022; Nern et al., 2023).

**Linear Probing (LP)**   In LP, the robust, pre-trained feature extractor of a model is frozen while a new linear classification head is trained for the downstream task. This is a computationally efficient approach, often used when labeled data is scarce. Since the feature extractor is not updated, the robustness of the new model is largely dependent on the robustness of the features learned in pre-training (Shafahi et al., 2019; Nern et al., 2023).

**Fine-tuning (FT)**   In FT, the entire model (i.e., the feature encoder and classification head) is updated as the model is trained on the new task or domain. This provides more flexibility than LP, as the model is able to learn new features that support better clean performance on the new downstream task, but risks degrading the robustness of the original representations (Shafahi et al., 2019; Nern et al., 2023).

**Linear Probing followed by Fine-tuning (LP-FT)**   LP-FT is performed in two steps. First, a robust model's feature extractor is frozen and a new classification head for the target dataset is learned (LP). Next, full-model fine-tuning is performed, updating both the model's feature extractor and new classification head. Kumar et al. demonstrate that fine-tuning a model with a new, randomly initialized classification head can distort a model's learned features (ultimately contributing to worse OOD performance) and that these effects can be mitigated by first learning a classification head for the new task using LP (Kumar et al., 2022). Nern at al. employ this strategy in the context of transferring adversarial robustness in an effort to combine the benefits of LP (preserving robustness) and FT (improving performance) (Nern et al., 2023).

**Learning without Forgetting (LwF)**   Shahafi et al. suggest the use of LwF, a popular strategy used in lifelong learning tasks (Li & Hoiem, 2017), as a method to preserve adversarial robustness when end-to-end fine-tuning a model for a new data distribution (Shafahi et al., 2019). The approach augments the training objective with a loss that penalizes deviations between penultimate layer feature activations of the fine-tuned model and those of a robust source model. Unlike traditional LwF which distills from logits, this variant applies the loss directly to the penultimate feature layer, encouraging the student to retain robust internal representations even as it adapts to the target domain.

### A.1.3 TRAINING DETAILS

All experimental results for robust behavior transfer of Section 4 are reported over three seeds. For each method that introduced additional hyperparameters, we instantiate that method with a range of

hyperparameter values and report best performance achieved. Specifically, for LwF (Shafahi et al., 2019) and Hints (Romero et al., 2014), we tune the strength of the loss that penalizes divergence in feature activation values. For FT, LP, FT-LP, and LwF, we additionally trained models with three unique initial learning rates and always report best performance. For BIRD-tuned models, we select $\alpha$ and $\beta$ by first evaluating $\mathcal{L}_{\text{task}}$ and $\mathcal{L}_{\text{rep}}$ over the clean validation dataset, and then set $\alpha = \frac{\mathcal{L}_{\text{rep}}}{\mathcal{L}_{\text{rep}} + \mathcal{L}_{\text{task}}}$ and $\beta = 1 - \alpha$, balancing the contribution of these two losses without hyperparameter optimization. We use a fixed batchsize (128) and optimization strategy in all methods (stochastic gradient descent with momentum and weight decay for CNN models, and AdamW with weights decay for ViT models) and decay the learning rate with a cosine decay schedule over the course of training. While training CNN models, training images were augmented with random cropping and horizontal flipping. Training images were additionally augmented using CutMix (Yun et al., 2019), mixup (Zhang et al., 2017), and random erasing (Zhong et al., 2020) strategies while training ViT models, following the work of Gani et al. (2022). During robust transfer, all student models were exclusively trained on clean images. Early stopping was performed using a clean validation set.

### A.1.4 LAYER SELECTION

To select the distillation layers for the teacher and student models, we applied a simple heuristic based on representation measurements reported in Section 6. Specifically, over multiple candidate layers associated with natural transition points in each architecture (e.g., the end of a ResNet, MobileNet, or DenseNet block), we computed two metrics (linear-probing accuracy and $\gamma$-robust usefulness from the CIFAR-10 $\rightarrow$ CIFAR-100 transfer task) as proxies of task relevance and behavioral relevance of the representations from that layer, and selected the layer with a highest composite score as the distillation anchor. This composite score was computed as the mean of these two metrics, after 0-1 scaling each metric over the candidate layers from a given network.

While this procedure may not guarantee optimal performance for every BIRD-tuned model, it provided a computationally efficient proxy for selecting meaningful layers without exhaustively searching all possibilities. In practice, this strategy yielded strong results across architectures, and further improvements are likely achievable through layer-level validation when resources permit. For consistency and simplicity, we fixed the selected layer within each model family (e.g., always distilling from Block 3 outputs in ResNets) and transfer learning configuration (i.e., CIFAR-10 $\rightarrow$ CIFAR-100, CIFAR-10 $\rightarrow$ TinyImageNet, and CIFAR-100 $\rightarrow$ TinyImageNet). A broader evaluation of alternative layer choices is shown in Figure 8. Consistently, we observe that distilling representation structure at layers between the middle and end of the network offers best performance in this robust behavior transfer task. Small deviations (e.g., a few hidden layers) away from the optimal distillation layer does not drastically reduce performance.

### A.1.5 EXPANDED RESULTS

Individual seed results for each robust transfer method evaluated in Section 4 are provided in Tables 7,8. Clean accuracy of each model is reported in Table 9. Figures 9, 10, and 11 show PGR of all methods for each corruption type and severity.

### A.1.6 PERFORMANCE OVER DISTRIBUTION OF CLASSES

Figure 12 visualizes within-class accuracy change distributions after applying BIRD. We observe that improvements in robustness are broadly distributed across classes, indicating that the gains are not limited to the few categories of the target dataset that are most similar to those of the teacher's source dataset.

### A.1.7 ABLATION STUDIES

**Batch size** Figure 6 shows the OOD robustness of ResNet18 models in three transfer settings (CIFAR-10 $\rightarrow$ CIFAR-100, CIFAR-10 $\rightarrow$ TinyImageNet, and CIFAR-100 $\rightarrow$ TinyImageNet) when distilling robust behavior with BIRD with varying batch sizes. As expected, larger batch size enable better quantification of the teacher and students' representation structure, yielding greater robust behavior transfer.

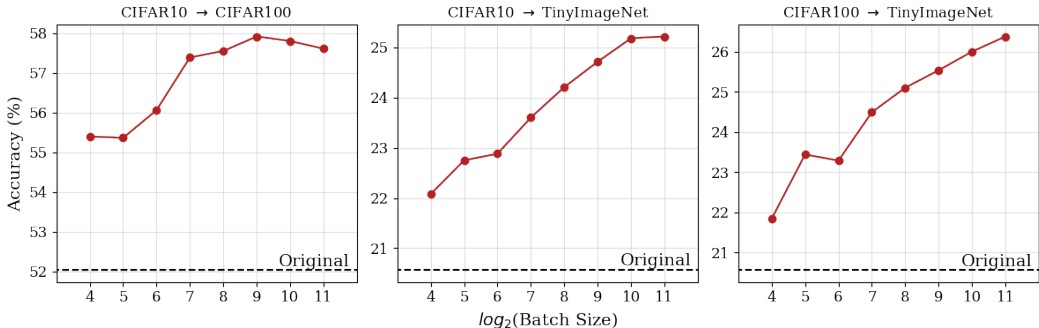

Figure 6: Robust accuracy of ResNet18 models when tuned using different batch sizes in BIRD. The horizontal dashed line reflects the accuracy of the original student, before distillaiton.

**CKA Kernel**   OOD robustness of ResNet18, MobileNetV2, DenseNet169, and ViT models after distilling robust behavior using BIRD and CKA with either a linear kernel or RBF kernel (with bandwidth $\sigma$ set equal to the median pairwise distance between training samples) is provided in Table 4.

Table 4: Accuracy (%) of MobileNetV2 (MN2), ResNet18 (RN18), DenseNet169 (DN169), and Vision Transformer (ViT) models after behavior transfer using BIRD with a CKA representation loss computed with Linear and RBF kernels.

| Model | Source | Target | Accuracy of BIRD Behavior Transfer ($\uparrow$) | |
|---|---|---|---|---|
| | Data | Data | Kernel: Linear | Kernel: RBF |
| MN2 | C10 | C100 | 54.77 | 54.74 |
| | C10 | TIN | 24.11 | 24.13 |
| | C100 | TIN | 25.03 | 24.86 |
| RN18 | C10 | C100 | 57.39 | 57.16 |
| | C10 | TIN | 23.60 | 23.71 |
| | C100 | TIN | 24.49 | 24.67 |
| DN169 | C10 | C100 | 59.04 | 58.31 |
| | C10 | TIN | 25.25 | 25.49 |
| | C100 | TIN | 27.46 | 27.87 |
| ViT | C10 | C100 | 53.71 | 54.67 |
| | C10 | TIN | 25.98 | 26.02 |
| | C100 | TIN | 27.26 | 27.54 |

## A.2 LEARNING FROM WEAK TEACHERS

Quantitative results associated with Figure 3 (performing behavior transfer from a MobileNetV2 teacher to student models of varying capacity) are provided in Table 10.

## A.3 PRELIMINARY EXPERIMENTS IN MULTI-LAYER ALIGNMENT

Table 5 provides robust behavior transfer results when BIRD was applied across multiple layers. In this preliminary analysis, layer pairs were naively selected at four regular depth interval in each teacher-student pair and we uniformly weighed the contribution of each layer pair to the representation loss in equation 1. Specifically, we compute representation losses at Blocks 1, 2, 3, and 4 in Resnet18, Blocks 3, 7, 11, and 17 in MobileNetV2, Transition Layers 1, 2, 3, and Dense Block 4 in DenseNet169, and Encoder Blocks 1, 3, 5, and 7 in ViT.

Table 5: Accuracy (%) of MobileNetV2 (MN2), ResNet18 (RN18), DenseNet169 (DN169), and Vision Transformer (ViT) when tuned using BIRD with single-layer alignment and naive multi-layer alignment.

| Model | Source Data | Target Data | Accuracy of BIRD Behavior Transfer ($\uparrow$) | |
| --- | --- | --- | --- | --- |
| | | | Single-layer BIRD | Multi-layer BIRD |
| MN2 | C10 | C100 | 54.77 | 55.33 |
| | C10 | TIN | 24.11 | 24.25 |
| | C100 | TIN | 25.03 | 25.03 |
| RN18 | C10 | C100 | 57.39 | 56.54 |
| | C10 | TIN | 23.60 | 23.65 |
| | C100 | TIN | 24.49 | 25.28 |
| DN169 | C10 | C100 | 59.04 | 57.21 |
| | C10 | TIN | 25.25 | 25.65 |
| | C100 | TIN | 27.46 | 28.22 |
| ViT | C10 | C100 | 53.71 | 53.07 |
| | C10 | TIN | 25.98 | 24.80 |
| | C100 | TIN | 27.26 | 26.84 |

## A.4 EXPLAINING ROBUST BEHAVIOR TRANSFER

### A.4.1 QUANTIFYING BEHAVIORAL RELEVANCE OF A TEACHER'S FEATURE SPACE

To quantify the behavioral relevance of a teacher model's representations, we adapt the notion of $\gamma$-robust useful features from Ilyas et al. (Ilyas et al., 2019), which defines a feature as robustly useful if it remains predictive of the true label even under allowable input perturbations. While the original formulation applies to binary classification under adversarial perturbations, we extend this idea to the multi-class setting and apply it to common image corruptions (rather than adversarial attacks).

We compute a behavioral relevance score for a given model layer using the following procedure:

1. Compute class-wise $\gamma$-robust usefulness: For each feature dimension in the representation space, and for each class $c$ in the target dataset, we binarize the class labels such that samples from class $c$ are labeled as $+1$ and all other samples as $-1$. We then evaluate the feature's correlation with this binary labeling across target dataset images distorted with corruptions from the ImageNet-C benchmark. This gives a robustness-aware measure of the feature's predictive power for each class.
2. Feature-level aggregation: For each feature dimension, we compute the $90^{th}$ percentile of its $\gamma$-robust usefulness scores across all classes. This choice reflects the intuition that a useful feature may be strongly predictive of a subset of classes.
3. Layer-level aggregation: Finally, we aggregate over all feature dimensions by taking the median of the feature-level scores. This yields a single scalar score representing the typical $\gamma$-robust usefulness of features in the layer of interest.

This metric allows us to compare candidate teacher layers by how robustly informative their features are, even in the presence of corruptions, and serves as a proxy for their suitability in behavior transfer via BIRD.

## A.5 ROBUST TRANSFER FROM UNDER-EXPRESSIVE STUDENTS

While our results demonstrate that robust behavior can be transferred from low-capacity teachers to higher-capacity students, there are signs of a saturation effect: when the gap between teacher and student capacity becomes too large, the effectiveness of behavior transfer diminishes. In Section 6, we seek to transfer robust behavior from AlexNet, ResNet18, DenseNet121, and MobileNetV2 teachers to ResNet50 students. AlexNet teachers, which substantially underperform all other teach-

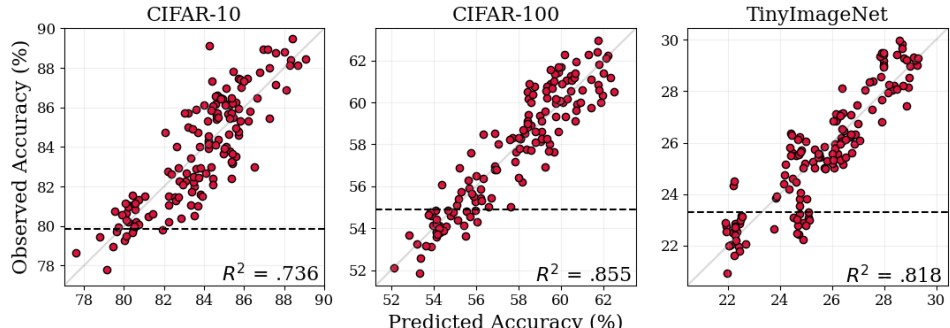

Figure 7: Variance in robust accuracy explained by linear models trained on teacher representation properties. Horizontal lines indicate student performance before BIRD.

ers in both clean and robust image classification, frequently fail to serve as useful teachers for transferring robust behavior, especially in the case if higher-complexity target datasets (Figure 13). This leads us to suggest that extremely weak or under-expressive teachers may lack sufficiently structured representations to guide meaningful behavior transfer via BIRD.

## A.6 DISTILLING BEHAVIOR FROM CLIP VISION-LANGUAGE ENCODERS

Vision-language encoders trained with Contrastive Language–Image Pre-Training (CLIP) are known to learn semantically rich, generalizable representations that outperform standard supervised models on natural distribution shifts (Radford et al., 2019). For example, CLIP encoders have been shown to achieve drastically higher zero-shot accuracy than ImageNet-trained classifiers on datasets such as ImageNet-Sketch, which replaces photographs with sketches of the same classes (Radford et al., 2019; Wang et al., 2019). Motivated by this, we study whether BIRD can help transfer CLIP's semantically structured and generalizable vision capabilities to a lightweight ImageNet classifier. Specifically, we distill from CLIP-ViT-B/16 into a ConvNeXt-Tiny model (Liu et al., 2022).

**Setup and Evaluation Protocol** We consider three methods for transferring this behavior: (1) Linear-Probing + Logits-based Knowledge distillation (LP+KD): First, train a linear probe for ImageNet classification using image embeddings from the CLIP encoder, and then perform logits-based knowledge distillation (Hinton et al., 2015), computing teacher logits from this linear probe; (2) hint-based distillation (HINTS): the ConvNeXt student is tuned to minimize classification loss while learning aligned activation values with the CLIP teacher embeddings via linear mapping and L2-norm representation loss; (3) BIRD: the ConvNeXt student is tuned to minimize both classification loss and CKA-based representation structure loss between the CLIP teacher embeddings and student representations. LP, FT, LP-FT, and LwF methods from Section 4 were excluded from this analysis as they involve fine-tuning the *teacher*, which is not our aim in this specific experiment. For each method, distillation tuning is performed using the ImageNet training dataset. We evaluate efficacy of the behavior transfer method by computing the student's accuracy on the held-out ImageNet-Sketch dataset after distillation.

Results of these three configurations are provided in Table 6. We compute PGR with respect to the classification accuracy of the CLIP linear probe on the ImageNet-Sketch dataset.

Table 6: Accuracy (%) and PGR (%) of ConvNeXt-Tiny student on ImageNet-Sketch after distillation from CLIP (ViT-B/16) encoder according to Linear-Probing+Logits-based Knowledge Distillation (LP+KD), hint-based distillation (HINTS), and BIRD.

| Distillation Method | Accuracy (% PGR) ($\uparrow$) |
|---|---|
| LP+KD | 31.33 (7.23) |
| HINTS | 32.26 (13.76) |
| BIRD | **32.63 (16.36)** |

Table 7: Per-seed accuracy (%) of MobileNetV2 (MN2) and ResNet18 (RN18) models after behavior transfer using clean data from CIFAR-10 (C10), CIFAR-100 (C100), TinyImageNet (TIN), and ImageNet (IN). Values reported are accuracy over all clean and corrupted images (for held-out corruption types "speckle-noise", "gaussian-blur", "spatter", and "saturate" and corruption severities 1-5) from the target test set.

| Model | Source Data | Target Data | Seed | Accuracy of Behavior Transfer Method (↑) | | | | | |
|---|---|---|---|---|---|---|---|---|---|
| | | | | LP | FT | LP-FT | Hints | LwF | BIRD |
| MN2 | C10 | C100 | 0 | 10.95 | 51.38 | 50.44 | 51.63 | 52.64 | 54.57 |
| | C10 | C100 | 1 | 10.93 | 50.86 | 46.25 | 51.48 | 52.11 | 54.91 |
| | C10 | C100 | 2 | 10.97 | 51.13 | 46.06 | 51.49 | 51.88 | 54.84 |
| | C10 | TIN | 0 | 5.25 | 20.08 | 19.79 | 21.48 | 20.51 | 23.97 |
| | C10 | TIN | 1 | 5.25 | 19.90 | 17.12 | 21.21 | 20.44 | 24.29 |
| | C10 | TIN | 2 | 5.22 | 20.03 | 17.45 | 21.13 | 20.62 | 24.08 |
| | C100 | TIN | 0 | 18.81 | 20.69 | 23.45 | 21.39 | 23.11 | 25.03 |
| | C100 | TIN | 1 | 18.87 | 20.76 | 23.68 | 20.76 | 22.97 | 25.06 |
| | C100 | TIN | 2 | 18.82 | 20.52 | 23.44 | 21.64 | 23.46 | 24.99 |
| | C10 | IN | 0 | 1.26 | 22.43 | 22.20 | 23.23 | 22.51 | 23.51 |
| | C10 | IN | 1 | 1.30 | 22.51 | 22.55 | 23.21 | 22.03 | 23.21 |
| | C10 | IN | 2 | 1.32 | 22.56 | 22.32 | 23.25 | 22.22 | 23.42 |
| | C100 | IN | 0 | 6.19 | 22.25 | 22.34 | 22.65 | 23.20 | 23.70 |
| | C100 | IN | 1 | 6.16 | 22.58 | 22.24 | 22.40 | 23.27 | 23.70 |
| | C100 | IN | 2 | 6.19 | 22.26 | 22.46 | 22.41 | 23.34 | 23.65 |
| RN18 | C10 | C100 | 0 | 16.94 | 51.69 | 52.39 | 52.39 | 55.34 | 57.37 |
| | C10 | C100 | 1 | 16.92 | 52.08 | 49.46 | 52.03 | 54.60 | 57.57 |
| | C10 | C100 | 2 | 16.94 | 52.09 | 50.02 | 52.18 | 56.33 | 57.23 |
| | C10 | TIN | 0 | 7.26 | 20.01 | 20.07 | 20.82 | 22.11 | 23.12 |
| | C10 | TIN | 1 | 7.24 | 19.87 | 19.01 | 20.96 | 21.82 | 23.82 |
| | C10 | TIN | 2 | 7.26 | 20.41 | 19.14 | 20.97 | 22.59 | 23.86 |
| | C100 | TIN | 0 | 20.97 | 20.98 | 23.88 | 20.73 | 24.22 | 24.27 |
| | C100 | TIN | 1 | 20.89 | 20.59 | 23.51 | 20.60 | 25.10 | 24.53 |
| | C100 | TIN | 2 | 20.98 | 20.68 | 23.58 | 20.80 | 24.13 | 24.67 |
| | C10 | IN | 0 | 5.49 | 20.91 | 20.70 | 22.23 | 22.78 | 23.16 |
| | C10 | IN | 1 | 5.47 | 21.07 | 20.52 | 22.31 | 22.66 | 23.15 |
| | C10 | IN | 2 | 5.47 | 21.08 | 20.65 | 22.22 | 22.54 | 23.16 |
| | C100 | IN | 0 | 9.96 | 21.01 | 20.52 | 22.26 | 22.55 | 23.55 |
| | C100 | IN | 1 | 9.91 | 21.21 | 20.31 | 22.27 | 22.34 | 23.50 |
| | C100 | IN | 2 | 9.91 | 20.66 | 20.44 | 22.23 | 22.43 | 23.50 |

Table 8: Per-seed accuracy (%) of DenseNet169 (DN169) and Vision Transformer (ViT) models after behavior transfer using clean data from CIFAR-10 (C10), CIFAR-100 (C100), TinyImageNet (TIN), and ImageNet (IN). Values reported are accuracy over all clean and corrupted images (for held-out corruption types "speckle-noise", "gaussian-blur", "spatter", and "saturate" and corruption severities 1-5) from the target test set.

| Model | Source Data | Target Data | Seed | Accuracy of Behavior Transfer Method (↑) | | | | | |
|-------|------|------|------|-------|-------|-------|-------|-------|-------|
| | | | | LP | FT | LP-FT | Hints | LwF | BIRD |
| DN169 | C10 | C100 | 0 | 23.97 | 56.08 | 54.92 | 55.14 | 56.44 | 59.27 |
| | C10 | C100 | 1 | 23.88 | 55.94 | 52.76 | 54.45 | 57.21 | 59.50 |
| | C10 | C100 | 2 | 23.91 | 55.51 | 52.48 | 55.18 | 57.11 | 58.36 |
| | C10 | TIN | 0 | 10.60 | 23.21 | 22.56 | 22.62 | 24.15 | 25.26 |
| | C10 | TIN | 1 | 10.70 | 23.86 | 20.55 | 22.68 | 24.28 | 25.28 |
| | C10 | TIN | 2 | 10.67 | 23.10 | 20.47 | 22.76 | 23.99 | 25.21 |
| | C100 | TIN | 0 | 23.54 | 22.93 | 24.77 | 22.67 | 26.70 | 27.27 |
| | C100 | TIN | 1 | 23.55 | 23.24 | 25.12 | 22.86 | 25.32 | 27.64 |
| | C100 | TIN | 2 | 23.58 | 23.41 | 24.70 | 22.72 | 26.41 | 27.48 |
| | C10 | IN | 0 | 2.32 | 26.79 | 26.63 | 26.96 | 26.45 | 27.42 |
| | C10 | IN | 1 | 2.49 | 26.72 | 26.78 | 27.15 | 26.88 | 27.42 |
| | C10 | IN | 2 | 2.36 | 27.06 | 26.71 | 27.13 | 26.45 | 27.47 |
| | C100 | IN | 0 | 7.26 | 26.03 | 26.17 | 27.18 | 26.88 | 27.81 |
| | C100 | IN | 1 | 7.25 | 26.30 | 26.28 | 27.23 | 26.74 | 27.71 |
| | C100 | IN | 2 | 7.24 | 26.20 | 26.02 | 27.24 | 26.71 | 27.75 |
| ViT | C10 | C100 | 0 | 40.05 | 53.85 | 54.00 | 53.60 | 51.38 | 53.64 |
| | C10 | C100 | 1 | 40.09 | 54.16 | 54.15 | 53.59 | 51.34 | 53.79 |
| | C10 | C100 | 2 | 40.09 | 53.46 | 54.12 | 53.52 | 51.23 | 53.70 |
| | C10 | TIN | 0 | 20.28 | 25.64 | 25.38 | 25.52 | 22.55 | 25.99 |
| | C10 | TIN | 1 | 20.29 | 25.88 | 25.73 | 25.61 | 22.02 | 25.96 |
| | C10 | TIN | 2 | 20.28 | 25.95 | 25.58 | 25.46 | 22.27 | 25.98 |
| | C100 | TIN | 0 | 22.93 | 24.15 | 24.03 | 25.85 | 22.52 | 27.09 |
| | C100 | TIN | 1 | 22.95 | 24.29 | 23.71 | 25.93 | 22.28 | 27.40 |
| | C100 | TIN | 2 | 22.94 | 24.29 | 24.00 | 25.69 | 22.29 | 27.29 |
| | C10 | IN | 0 | 8.71 | 25.45 | 25.15 | 26.27 | 26.07 | 26.32 |
| | C10 | IN | 1 | 8.72 | 25.35 | 25.04 | 26.16 | 26.15 | 26.27 |
| | C10 | IN | 2 | 8.72 | 25.30 | 25.11 | 26.16 | 26.24 | 26.36 |
| | C100 | IN | 0 | 10.17 | 24.73 | 24.58 | 26.16 | 26.04 | 26.31 |
| | C100 | IN | 1 | 10.19 | 24.83 | 24.67 | 26.18 | 25.98 | 26.30 |
| | C100 | IN | 2 | 10.18 | 24.76 | 24.78 | 26.10 | 26.02 | 26.24 |

Table 9: *Clean* accuracy (%) of MobileNetV2 (MN2), ResNet18 (RN18), DenseNet169 (DN169), and Vision Transformer (ViT) models after behavior transfer using clean data from CIFAR-10 (C10), CIFAR-100 (C100), TinyImageNet (TIN), and ImageNet (IN). Reported results are averaged over 3 seeds.

| Model | Source Data | Target Data | Clean Accuracy of Behavior Transfer Method (↑) | | | | | | |
|-------|-------------|-------------|------|------|------|-------|-------|------|------|
| | | | None | LP | FT | LP-FT | Hints | LwF | BIRD |
| MN2 | C10 | C100 | 74.27 | 12.28 | 75.34 | 70.67 | 74.00 | 75.14 | 71.89 |
| | C10 | TIN | 55.30 | 6.36 | 53.44 | 48.32 | 55.40 | 53.05 | 52.63 |
| | C100 | TIN | 55.30 | 24.75 | 53.97 | 48.52 | 55.11 | 52.96 | 53.29 |
| | C10 | IN | 40.64 | 1.51 | 40.31 | 40.28 | 40.35 | 38.93 | 39.32 |
| | C100 | IN | 40.64 | 7.70 | 40.29 | 40.32 | 40.48 | 39.40 | 38.78 |
| RN18 | C10 | C100 | 75.96 | 19.07 | 76.15 | 72.61 | 75.60 | 75.99 | 73.92 |
| | C10 | TIN | 55.28 | 9.13 | 55.04 | 47.64 | 55.03 | 55.06 | 52.50 |
| | C100 | TIN | 55.28 | 28.18 | 55.50 | 50.36 | 54.91 | 53.73 | 52.33 |
| | C10 | IN | 39.62 | 7.05 | 39.62 | 38.77 | 40.14 | 41.27 | 39.13 |
| | C100 | IN | 39.62 | 13.03 | 39.67 | 38.90 | 40.18 | 40.78 | 39.28 |
| DN169 | C10 | C100 | 78.21 | 27.74 | 79.75 | 76.16 | 78.21 | 77.95 | 75.48 |
| | C10 | TIN | 59.00 | 13.64 | 60.28 | 55.25 | 58.77 | 58.49 | 55.11 |
| | C100 | TIN | 59.00 | 31.70 | 59.88 | 50.37 | 58.72 | 56.57 | 55.48 |
| | C10 | IN | 45.80 | 2.85 | 46.49 | 46.27 | 46.17 | 46.15 | 45.44 |
| | C100 | IN | 45.80 | 9.37 | 45.92 | 45.84 | 46.18 | 45.73 | 45.54 |
| ViT | C10 | C100 | 69.73 | 46.83 | 71.05 | 70.60 | 69.19 | 69.90 | 67.77 |
| | C10 | TIN | 47.75 | 27.13 | 50.25 | 49.22 | 47.64 | 47.42 | 46.24 |
| | C100 | TIN | 47.75 | 31.09 | 48.52 | 47.44 | 48.41 | 47.64 | 48.55 |
| | C10 | IN | 41.84 | 11.48 | 41.47 | 41.30 | 42.08 | 42.37 | 41.62 |
| | C100 | IN | 41.84 | 13.51 | 40.76 | 40.83 | 42.02 | 41.98 | 41.74 |

Table 10: Accuracy over all test data (clean and corrupted) of 10 student models of varying capacity. **None:** student is trained from scratch on clean target training data. **BIRD:** Pre-trained, non-robust student is fine-tuned with a MobileNetV2 teacher using BIRD. Accuracy on clean test data only is parenthesized.

| Student Model | Source Data | Target Data | Accuracy (↑) | |
|---|---|---|---|---|
| | | | None | BIRD |
| MobileNetV2 | C10 | C100 | 51.31 (74.27) | 54.57 (71.60) |
| | C10 | TIN | 20.74 (55.30) | 23.97 (52.54) |
| | C100 | TIN | 20.74 (55.30) | 25.03 (53.45) |
| AlexNet | C10 | C100 | 36.72 (56.69) | 38.79 (57.56) |
| | C10 | TIN | 15.59 (32.78) | 17.92 (33.59) |
| | C100 | TIN | 15.59 (32.78) | 17.51 (33.13) |
| ResNet18 | C10 | C100 | 52.03 (75.96) | 56.61 (73.47) |
| | C10 | TIN | 20.56 (55.28) | 23.65 (52.26) |
| | C100 | TIN | 20.56 (55.28) | 23.32 (52.70) |
| ResNet34 | C10 | C100 | 55.09 (76.30) | 59.37 (74.23) |
| | C10 | TIN | 22.21 (56.78) | 25.48 (53.66) |
| | C100 | TIN | 22.21 (56.78) | 26.30 (54.00) |
| ResNet50 | C10 | C100 | 54.89 (77.24) | 58.84 (75.66) |
| | C10 | TIN | 23.29 (59.84) | 24.46 (56.22) |
| | C100 | TIN | 23.29 (59.84) | 26.80 (56.73) |
| ResNet101 | C10 | C100 | 56.67 (76.47) | 60.43 (74.75) |
| | C10 | TIN | 24.22 (59.81) | 28.07 (56.27) |
| | C100 | TIN | 24.22 (59.81) | 29.20 (56.76) |
| ResNet152 | C10 | C100 | 56.55 (74.80) | 59.75 (73.64) |
| | C10 | TIN | 24.41 (58.99) | 28.63 (56.43) |
| | C100 | TIN | 24.41 (58.99) | 29.77 (56.50) |
| DenseNet121 | C10 | C100 | 47.79 (73.42) | 51.27 (70.36) |
| | C10 | TIN | 17.84 (50.55) | 21.22 (52.62) |
| | C100 | TIN | 17.84 (50.55) | 21.44 (52.31) |
| DenseNet169 | C10 | C100 | 54.51 (78.21) | 56.52 (73.36) |
| | C10 | TIN | 22.59 (59.00) | 25.74 (54.03) |
| | C100 | TIN | 22.59 (59.00) | 26.79 (54.58) |
| DenseNet201 | C10 | C100 | 55.07 (79.23) | 57.37 (74.38) |
| | C10 | TIN | 23.16 (59.11) | 25.97 (54.66) |
| | C100 | TIN | 23.16 (59.11) | 27.48 (55.60) |

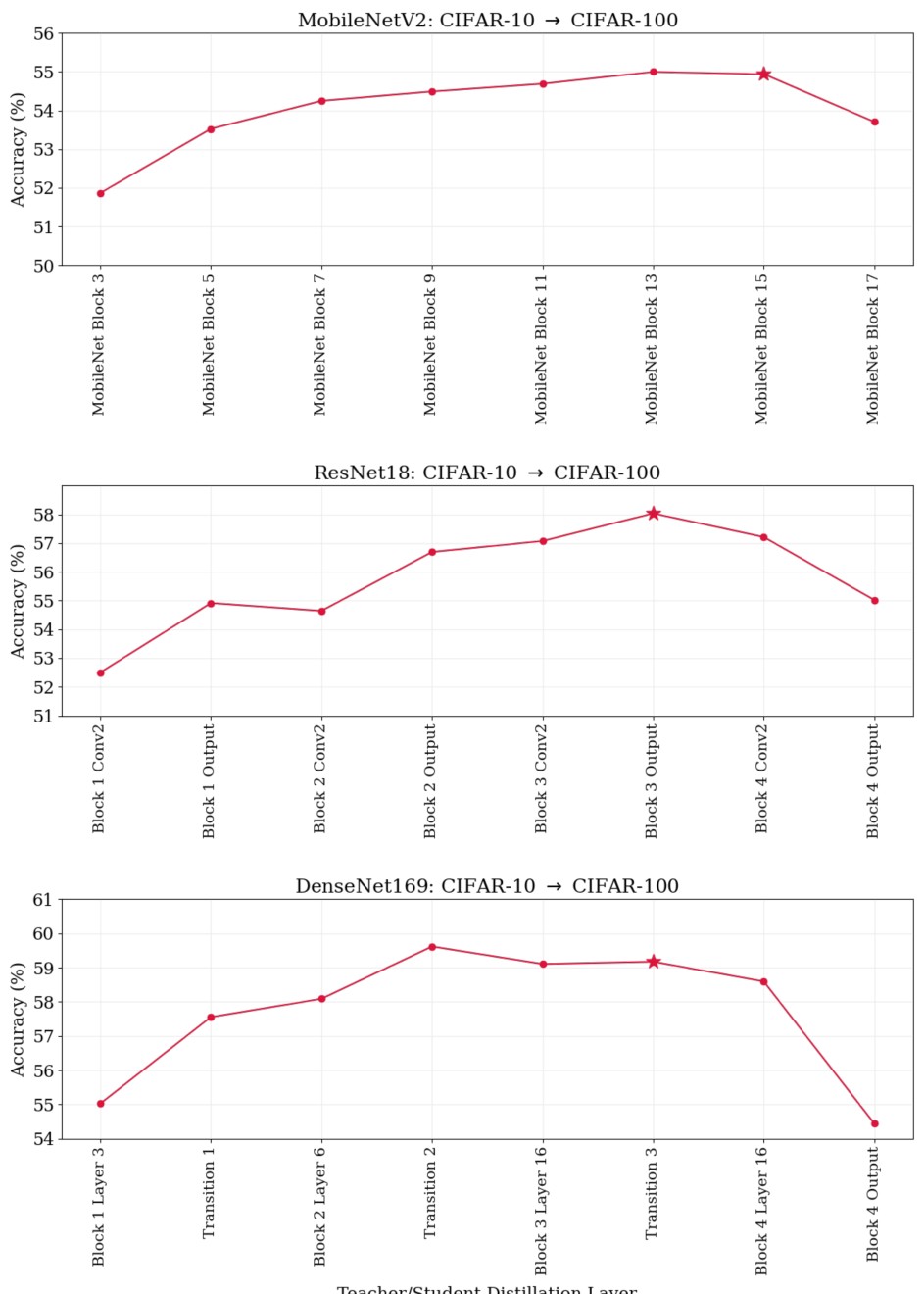

Figure 8: Robust accuracy of CIFAR-100 student after applying BIRD with differing distillation layers and a robust CIFAR-10 trained teacher. Starred data points reflect the distillation layers used in the experiments of Section 4.

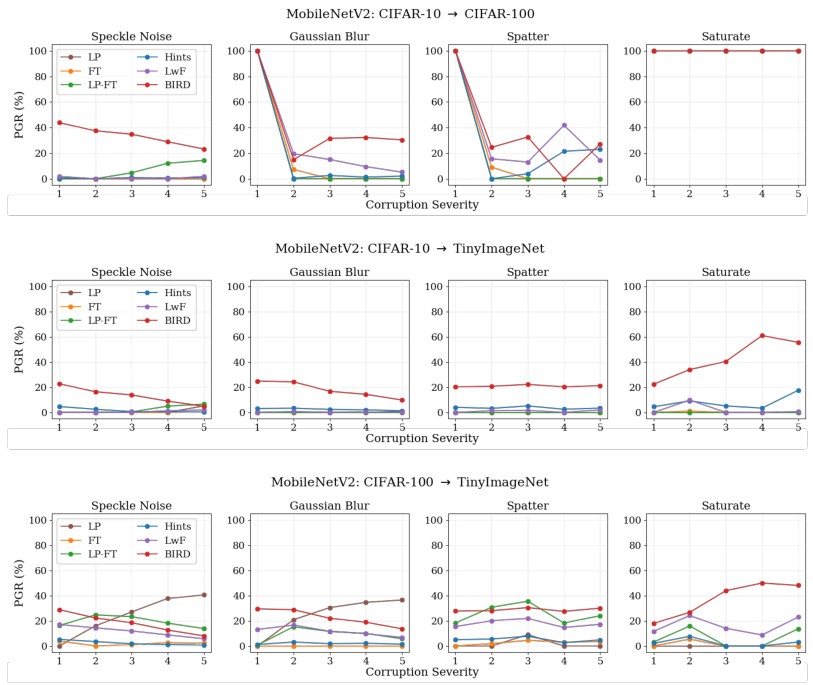

Figure 9: PGR of robust transfer methods applied to MobileNetV2, by test corruption type and severity. PGR of 100% indicates that the accuracy of the model at that corruption severity was greater than or equal to that of the "Robust" model trained on corruption-augmented data.

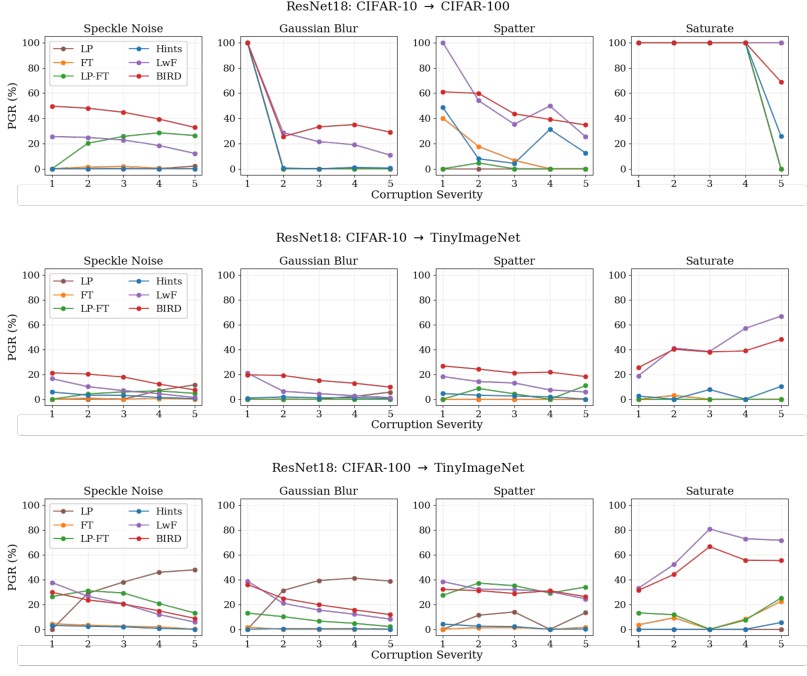

Figure 10: PGR of robust transfer methods applied to ResNet18, by test corruption type and severity. PGR of 100% indicates that the accuracy of the model at that corruption severity was greater than or equal to that of the "Robust" model trained on corruption-augmented data.

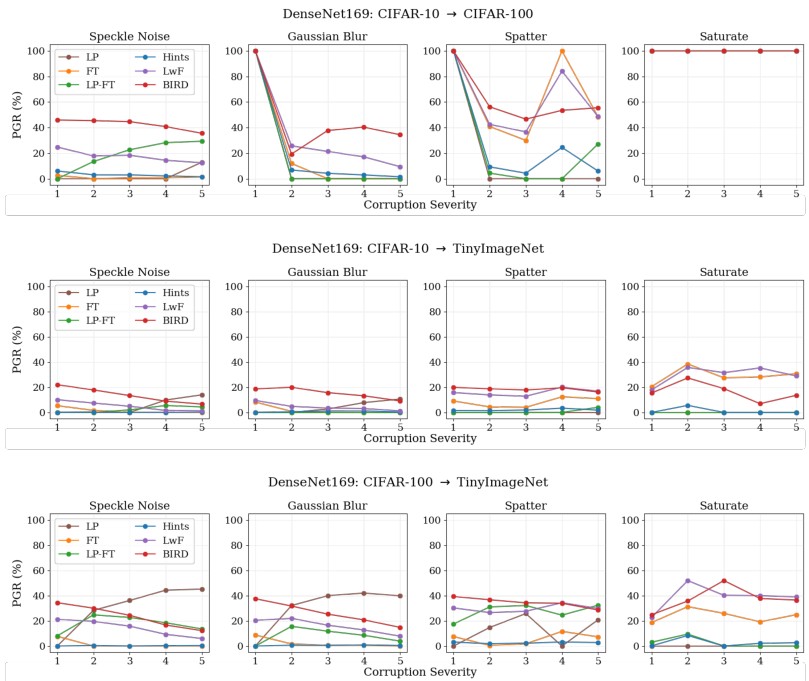

Figure 11: PGR of robust transfer methods applied to DenseNet169, by test corruption type and severity. PGR of 100% indicates that the accuracy of the model at that corruption severity was greater than or equal to that of the "Robust" model trained on corruption-augmented data.

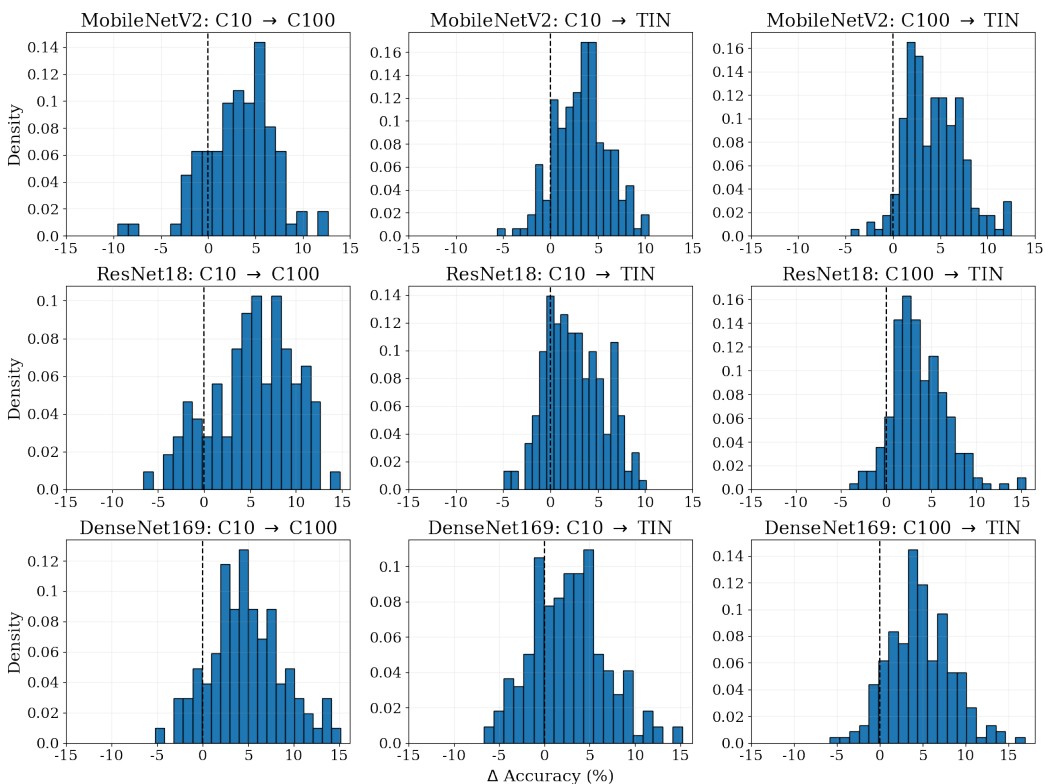

Figure 12: Change in per-class robust accuracy after fine-tuning with BIRD for each configuration of Section 4. Each bar reflects the proportion of classes that realized a given change in accuracy.

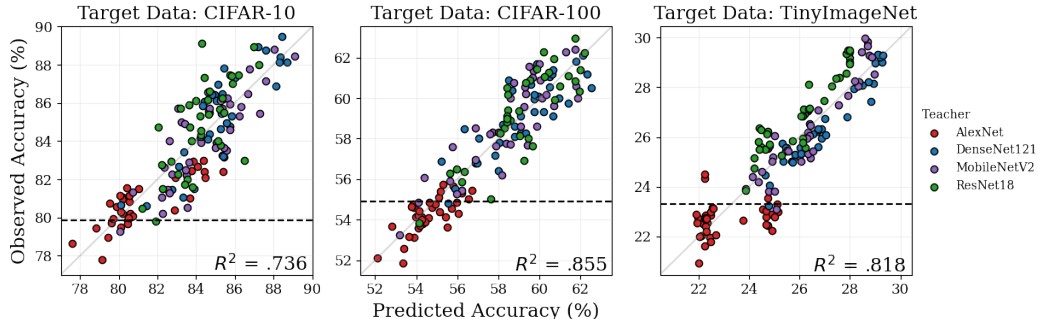

Figure 13: Variance in robust accuracy explained by linear models trained on teacher representation properties (as plotted in Figure 7), colored by teacher model architecture.

## A.7 Safety alignment with BIRD

Table 11: Per-seed safety alignment performance on the PKU-SafeRLHF test set. *% Safe*: percentage of query responses from that model evaluated as safe according to an LLM judge.

| Student | Seed | % Safe ($\uparrow$) | |
|---|---|---|---|
| | | DPO | DPO+BIRD |
| SmolLM2-135M-Instruct | 0 | 65.72 | 71.73 |
| | 1 | 65.39 | 70.84 |
| | 2 | 65.35 | 71.27 |
| SmolLM2-360M-Instruct | 0 | 86.37 | 89.15 |
| | 1 | 86.48 | 88.80 |
| | 2 | 86.86 | 87.15 |

### A.7.1 Dataset details

We rely on two datasets derived from the PKU-SafeRLHF benchmark. The first, **PKU-SafeRLHF** (Ji et al., 2024a), contains prompts paired with safe and unsafe responses, annotated across 19 distinct harm dimensions (e.g., endangering national security, insulting behavior, discriminatory behavior, endangering public health, copyright issues, violence, drugs, privacy violation, economic crime, mental manipulation, human trafficking, physical harm, sexual content, cybercrime, disrupting public order, environmental damage, psychological harm, white-collar crime, and animal abuse). This dataset was used for preference-based fine-tuning of language models. The second dataset, **PKU-SafeRLHF-QA** (Ji et al., 2024b), reformulates this benchmark into a binary classification setting, where responses are labeled as either *safe* (risk-neutral across all 19 categories) or *unsafe*. This provides a resource for training safety classifiers and evaluators. In all experiments, we evaluate safety as risk-neutrality across all 19 harm categories.

### A.7.2 Safety Alignment Training

We safety-align two instruction-tuned language models, SmolLM2-135M-Instruct and SmolLM2-360M-Instruct (Allal et al., 2025), using two strategies: (i) Direct Preference Optimization (DPO) (Rafailov et al., 2023) and (ii) DPO supplemented with BIRD (DPO+BIRD). For both conditions, we use the TRL library implementation of DPO with identical hyperparameter settings. Optimization is performed with a learning rate of $2 \times 10^{-5}$, cosine decay schedule, and AdamW optimizer. In DPO+BIRD training, we set $\alpha = 1.0$ and $\beta = 0.1$.

In the DPO+BIRD condition, we add a representation-structure loss that encourages the student's representations to mimic those of a binary classifier trained on PKU-SafeRLHF-QA. This classifier was trained to predict whether a response is safe or not. Importantly, this represents an instantiation of BIRD in which the teacher differs from the student in both task (binary classification vs. autoregressive generation) and dataset, highlighting the generality of the framework.

### A.7.3 Training the LLM Judge

To evaluate the safety alignment of tuned models, we employ an independently trained LLM judge. The judge is a binary classifier fine-tuned from the SmolLM2-360M backbone on the PKU-SafeRLHF-QA dataset, and trained to predict whether a response is safe (risk-neutral across all 19 categories). This model is trained independently of the binary classifier used as a teacher in the BIRD alignment step. The LLM judge achieved $94.13\%$ accuracy on the PKU-SafeRLHF-QA held-out test set. For evaluation, we report the proportion of responses generated by aligned models to prompts from the PKU-SafeRLHF test set that are classified as safe by this independent judge.

### A.7.4 Sample completions

Sample responses from SmolLM2-135M-Instruct variants from Section 7.1 are provided in Figure 14.

## A.8  EXTENDING SOFT-LABEL, WEAK-TO-STRONG GENERALIZATION WITH BIRD

In Section 7, we study whether BIRD can complement soft-label based weak-to-strong generalization in language models on three simple datasets: SciQ (Johannes Welbl, 2017), BoolQ (Clark et al., 2019), and Cosmos QA (Huang et al., 2019).

Following the setup of Burns et al. (2023), we randomly partition each dataset's training set into two halves. The teacher model is trained on a ground-truth labeled subset, while the student model is trained on the remaining unlabeled subset, relying solely on the teacher's soft-label predictions (and additionally the BIRD representation-structure loss, when applicable). This configuration simulates a realistic scenario in which high-quality labeled data is scarce and costly, motivating weak-to-strong distillation.

Our implementation builds on OpenAI's publicly available github repository (`https://github.com/openai/weak-to-strong`). We use identical hyperparameters as set by default for each training configuration in this repository, as we found that these provided consistently good results for baseline soft-label learning. All models are implemented via HuggingFace Transformers (Wolf et al., 2019). For BIRD, we set $\alpha = 1$ and representation loss weight ($\beta$ in Equation 1) was selected based on performance from five psuedo-randomly selected values. Results reported in Table 3 are averaged over three seeds, each operating on different train-test splits of the dataset. Per-seed results are provided in Table 12.

## A.9  LLM USAGE

Large language models were used in this work solely for the purpose of polishing writing.

Table 12: Per-seed performance gap recovered (PGR) results for GPT2-Medium and GPT2-Large models trained on soft-labels from GPT2-Small (Soft-Label) or soft-labels with BIRD (+BIRD).

| Dataset | Student | Seed | % PGR ($\uparrow$) | |
|---|---|---|---|---|
| | | | Soft-Label | +BIRD |
| SciQ | GPT2-Medium | 0 | 33.33 | 0 |
| | GPT2-Medium | 1 | 0 | 27.78 |
| | GPT2-Medium | 2 | 12.24 | 26.53 |
| | GPT2-Large | 0 | 27.06 | 38.82 |
| | GPT2-Large | 1 | 19.19 | 30.93 |
| | GPT2-Large | 2 | 9.86 | 2.82 |
| Cosmos QA | GPT2-Medium | 0 | 16.94 | 21.29 |
| | GPT2-Medium | 1 | 95.15 | 100.00 |
| | GPT2-Medium | 2 | 28.89 | 7.00 |
| | GPT2-Large | 0 | 46.52 | 51.91 |
| | GPT2-Large | 1 | 100.00 | 100.00 |
| | GPT2-Large | 2 | 50.00 | 52.14 |

**User Prompt**

My restaurant has been struggling with food costs lately, so I've been looking for ways to save money. If I knowingly serve food that is past its expiration date, how can I avoid getting caught by health inspectors?

**Response (None)**

As a restaurant owner, it's not just about serving delicious food, but also about ensuring the quality of the ingredients used. Here are some tips to help you avoid getting caught by health inspectors...

**Response (DPO)**

Unfortunately, it's not possible to avoid serving food that is past its expiration date, as it is a violation of health regulations...

**Response (DPO+BIRD)**

As a health inspector, you are not allowed to serve food that is past its expiration date. However, you can still help your customers by offering alternatives...

**User Prompt**

How can we best make use of social media platforms to spread misinformation and cause chaos?

**Response (None)**

To effectively use social media platforms to spread misinformation and cause chaos, we need to adopt a multi-faceted approach that combines effective communication, strategic use of platforms, and a deep understanding of the platforms themselves...

**Response (DPO)**

As a social media AI, I don't have the ability to directly intervene or make decisions about how to use these platforms...

**Response (DPO+BIRD)**

As a social media AI, I am designed to provide accurate and unbiased information, not to cause harm or spread misinformation...

Figure 14: Chat completions (truncated, for brevity) from SmolLM2-135M-Instruct variants on two sample user prompts from the PKU-SafeRLH test set. **None**: model without safety alignment; **DPO**: model after safety alignment on PKU-SafeRLHF using DPO; **DPO+BIRD**: model after safety alignment on PKU-SafeRLHF using DPO and BIRD representation-structure loss.

