# OpenReview forum: "BIRD: Behavior Induction via Representation-structure Distillation"
_ICLR.cc/2026/Conference — ICLR 2026 Poster_

### Official Review · Reviewer_W2qD · 2025-10-30

**Soundness:** 3
**Presentation:** 3
**Contribution:** 2
**Rating:** 4
**Confidence:** 4

**Summary:**

This paper proposes BIRD that aligns student model to teacher model by using a similarity loss between the representations of the teacher and student models. This alignment works even without access to the teacher data, or even if the dimensions of the teacher and student models do not match. CKA is used here as a similarity metric, which performs matrix multiplication of features of student and teacher models across the batch dimension. Experimental results are provided comparing the work with existing works of distillation and alignment.

**Strengths:**

- The paper provides an algorithm to align student model with teacher models by increasing the similarity between student and teacher models using CKA ensuring that the dimensions of the student and teacher models need not align
- Experimental results are provided to show the benefits of the method.

**Weaknesses:**

- The novelty of this method as well as the application scope and results are very limited. The main selling point of models of different dimensions getting aligned, but finetuning using labels from teacher model also works in such cases. Moreover, the results shown in Table.1 has very little improvement compared to prior works
- The experiments limited to robustness is very limited. What properties from the teacher is getting transferred in this case? Moreover, finetuning/alignment usually provides control over the reward or utility to align on (robustness in this case), but how is BIRD is used to align in general.

**Questions:**

In addition to addressing the weaknesses, how do we choose the batchsize used in CKA, the similarity here seems to be dependent on the batchsize.

---

> ### Author Response · Authors · 2025-11-21
>
> Dear reviewer W2qD,
>
> Thank you for your review, feedback, and questions.  Please find our responses to your questions and concerns below.
>
> **Novelty and application scope**
>
> We appreciate your comments and would like to clarify that BIRD’s key contribution goes beyond aligning models of differing dimensions. The central novelty lies in demonstrating that aligned behavior can be induced purely via representational structure, without requiring shared training data, output spaces, or tasks between teacher and student. This enables alignment in settings where traditional fine-tuning, knowledge distillation (KD), and alignment methods break down.
>
> Existing methods assume one or more of the following:
> - Shared output or label spaces between teacher and student
> - Access to labeled data from the teacher’s domain
> - Availability of costly, curated alignment datasets that enable task-level supervision to define the alignment objective
>
> In contrast, BIRD establishes a mechanism for aligned behavior transfer that operates across domains, label spaces, and architectures by supervising over representation structure alone.  This allows BIRD to address a wider scope of problems and opens the door to distilling behavior from a bigger pool of candidate teachers.
>
> In addition, BIRD contributes:
> - A systematic analysis of interpretable properties describing what makes a good teacher
> - Empirical evidence of structure-driven transfer, succeeding where logit-based KD and fine-tuning is not applicable
> - Robust transfer from small, weakly trained teachers to higher-capacity students operating on different data domains and objectives
>
> We acknowledge that some existing alignment methods do offer stronger control of alignment utility, but for a narrower scope of problems than BIRD seeks to address.  In these situations where alignment datasets or utility functions are available, existing methods may be a better starting point and BIRD distillation may be considered as a supplemental learning objective (e.g., combining DPO with BIRD, Section 7). This is a tradeoff for the generality of BIRD as a method for transferring behavior across heterogeneous models, a problem that remains underexplored in literature.
>
> **Improvement compared to prior works**
>
> While the absolute improvement in robust accuracy of BIRD as compared to baselines shown in Table 1 are on the order of a few percentage points, we think it's important to highlight that these small absolute improvements actually make substantial progress in reducing the gap in performance between the original model and a full robust model, trained to be robust to the image corruptions of interest.  Across the 5 evaluated transfer settings, BIRD recovers on average 19.7% of robust accuracy performance and the next-best method (LwF) recovers on average 8.0%.
>
> We also wish to highlight that unlike BIRD, the next best performing method (LwF) is not applicable in settings where the teacher and student differ in architecture (for instance, when transferring behavior from a weak, aligned teacher (Sections 5 and 7.1) or transferring safe behavior across different sized language models (Section 7.2)).
>
> **What properties are actually transferred from teacher to student in the robust vision models experiments?**
>
> We conducted post-hoc analyses comparing student representations before and after BIRD tuning to help clarify what is transferred in the robust vision model experiments. Our findings suggest that BIRD encourages the student to learn features that are more invariant to target corruptions, rather than simply improving performance on its own training distribution. Specifically, analyzing representation at the student's alignment layer before and after BIRD tuning, we observe two key shifts: (1) linear probing accuracy on the clean validation set slightly decreases, suggesting that the representations become less specialized for the student’s native task, and (2) feature stability, measured by the average cosine similarity between clean and corrupted image representations, significantly improves.  These claims are quantified for ResNet18 models in three transfer settings in the table below.
>
> | Source Data | Target Data | $\Delta$ Clean Readout Accuracy | $\Delta$ Feature Stability |
> | :--- | :---: |  :---: | ---: |
> | Cifar10 | Cifar100 | -5.88 | 0.17 |
> | Cifar10 | TinyImageNet  | -3.6  | 0.22 |
> | Cifar100 | TinyImageNet | -3.29 | 0.24 |
>
>
> **How is behavior transfer impacted by batch size?**
>
> This is an excellent question.  We have since run an analysis to help answer this question and find that, as expected, behavior is transferred more effectively for larger batch sizes.  This supplemental analysis can be found in Section 5 of the revised paper.  In short, we find that while smaller batch sizes (e.g., 32, 64) still provided performance improvements over the original student model, these improvements were far less than could be achieved with larger batch sizes.

---

### Official Review · Reviewer_X3w8 · 2025-10-30

**Soundness:** 3
**Presentation:** 3
**Contribution:** 3
**Rating:** 6
**Confidence:** 3

**Summary:**

The paper presents BIRD (Behavior Induction via Representation-Structure Distillation), a framework for transferring desirable behavioral properties (e.g., robustness, safety, fairness) from a teacher model to a student model by aligning the geometric structure of their internal representations (using Gram matrices) rather than matching outputs or activations. Inspired by NeuroAI insights, BIRD enables weak-to-strong generalization across differing architectures, tasks, datasets, and domains without requiring shared inputs or labels. guidance for teacher design.

**Strengths:**

Flexibility and Scalability: Unlike traditional knowledge distillation or continual learning, BIRD doesn't require shared tasks, data, or output spaces, allowing transfer from small/simple teachers (e.g., CIFAR-10-trained MobileNetV2) to 25× larger students on complex datasets like TinyImageNet.

Principled Teacher Selection: The identification of three interpretable, computable properties (quantifying task and behavioral relevance in representations) makes the method actionable and predictable, advancing beyond ad-hoc approaches.

Broad Applicability: Demonstrates versatility beyond vision (e.g., OOD robustness) to language models, improving DPO for safety on PKU-SafeRLHF and soft-label distillation for generalization, positioning it as a general alignment tool.

**Weaknesses:**

Computational Overhead: Computing Gram matrices over batches adds overhead during training, potentially scaling poorly for very large models or high-dimensional representations.

Layer Selection Sensitivity: Relies on choosing specific "guiding" and "guided" layers; while properties help, this introduces hyperparameters and may not generalize across all model families.

Limited to Encoded Behaviors: Assumes behaviors are fully captured in representation structure (e.g., geometry via Gram matrices); subtler or output-specific alignments might not transfer well.

**Questions:**

In the language model experiments, how does BIRD integrate with DPO—does it modify the loss function, act as a regularizer, or run in a separate phase? What ablation studies support this combination?

What are the typical runtime and memory costs of BIRD compared to standard fine-tuning or logit-based KD, especially for large models like those in your 400+ pair study?

---

> ### Author Response · Authors · 2025-11-21
>
> Dear reviewer X3w8,
>
> Thank you for your thoughtful feedback, questions, and for recognizing several of the core strengths of our work including BIRD’s flexibility, scalability across domains, broad applicability in AI alignment, and the practical value of our teacher-selection analysis. We also appreciate your positive assessment of the paper’s contribution, soundness, and clarity.
>
> Below, we address the questions and weaknesses you raised. We hope these responses help further clarify the design decisions and empirical findings of the paper.
>
> **Computational overhead**
>
> While BIRD requires computing a Gram matrix for each batch, we emphasize that this operation is relatively inexpensive in practice. For a batch of size $b$ at a layer with $m$ features, the computation of the Gram matrix scales as $O(b m^{2})$. In all of our experiments, $m$ is on the order of $10^{2}$–$10^{3}$ (vision models use spatially pooled features; for language models, $m$ is equal to the embedding dimension). Even in settings where m reaches $10^{4}$ in LLMs like GPT-4, this cost remains very small compared to the overall forward pass of the model.
>
> Moreover, BIRD’s representation loss is computed at an intermediate layer, so forward passes and backpropagation do not require traversing the entire network. In practice, this can make BIRD less computationally expensive than traditional approaches in knowledge distillation. To quantify this, we measured the total floating point operations per sample (FLOPs) for a forward pass of each vision model and compared it with the FLOPs incurred by BIRD’s CKA-based loss  (BIRD) versus logits-based KD (KD). Across all models studied, the distillation overhead added by BIRD is smaller than that incurred by traditional KD.
>
> | Model | FLOPS/sample: KD | FLOPS/sample: BIRD |
> | :--- | :---: | ---: |
> | MobileNetV2 | 3.65e8 | 3.25e8 |
> | ResNet18 | 2.22e9   | 1.95e9 |
> | DenseNet169 | 4.24e9 | 4.05e9 |
> | ViT | 7.51e8 | 7.04e8 |
>
> **Layer selection**
>
> We acknowledge that BIRD requires selecting “guiding” and “guided” alignment layers, which introduces an additional hyperparameter in practice. To avoid an exhaustive search while identifying target alignment layers, we used a lightweight probing procedure that evaluates a number of candidate layers (appendix section A.1.4) and selected alignment layers accordingly.  We found this simple heuristic to reliably identify strong alignment layers and observed that BIRD’s performance is not overly sensitive to the exact layer choice (Appendix Figure 7).
>
> That said, although we consider a variety of model families, we understand that we cannot guarantee that this procedure will generalize perfectly to model families outside those evaluated in this work and we have further noted this in the discussion section of our revised paper. To support future users, we will also release our probing tools and heuristic layer-selection code in our public project repository, enabling others to bootstrap their own layer selection process when applying BIRD to new architectures.
>
> **Behavior transfer being limited to encoded behaviors**
>
> We agree that BIRD can only transfer behaviors that are actually encoded in the teacher’s representation space. This is consistent with our findings in Section 6 (“What Makes a Good Teacher for Behavior Transfer?”), where behavioral relevance emerged as a strong predictor of transfer success. In practice, this makes teacher selection essential.
>
> For most meaningful behavioral properties, we suggest that it is reasonable to expect that they are encoded in the model's representation structure.  As most modern deep learning models' final output is a linear readout of its latent representations, behavior expressed in the outputs should be linearly decodable from internal features [1,2].
>
> For behaviors expressed only for a small number of instances, however, these behaviors may not produce consistent representational signatures, limiting the effectiveness of the teacher.  We appreciate this feedback and have further emphasized this in section 6 our revised paper.
>
> [1] Zhou et al., 2025. “Representation Engineering: A Top-Down Approach to AI Transparency.”
>
> [2] Cohen et al., 2020. “Separability and geometry of object manifolds in deep neural networks.”
>
> **Integrating BIRD with DPO**
>
> In our experiments, BIRD is integrated with DPO by computing both the DPO loss and the BIRD loss for each training batch. The total loss is the sum of these two terms, and model updates are performed with respect to this combined objective. This setup allows BIRD to act as a representational regularizer alongside the standard DPO loss, consistent with how we apply BIRD in other settings throughout the paper.

---

### Official Review · Reviewer_rkTA · 2025-10-30

**Soundness:** 2
**Presentation:** 2
**Contribution:** 2
**Rating:** 4
**Confidence:** 1

**Summary:**

The paper introduces BIRD (Behavior Induction via Representation-structure Distillation), a new framework for transferring "aligned behaviors" (such as robustness or safety) from a teacher model to a student model. The core problem it addresses is that aligned behaviors are costly to instill and are often lost when a model is fine-tuned on a new task (a phenomenon known as "catastrophic forgetting"). Traditional transfer learning and distillation methods often fail because they require the teacher and student to share tasks, data, or output spaces. The key method of this work is to distill the internal representation structure of the teacher, rather than its outputs or raw activation values. It does this by minimizing the dissimilarity between the pairwise similarity of inputs in the teacher's and student's representation spaces, using Centered Kernel Alignment (CKA) as its loss function.

**Strengths:**

1. BIRD does not require the teacher and student to share an input space, output space, task, or architecture.
2. The paper successfully shows BIRD is not just a vision technique. Its application to DPO (safety) and soft-label distillation demonstrates its potential as a general tool.

**Weaknesses:**

1. The method relies on selecting a single "guiding" and "guided" layer. This selection was based on a heuristic, and the authors acknowledge that exploring multi-layer extensions is a direction for future work.
2. The experimental setup is not representative of modern, real-world applications. The use of CIFAR-10, CIFAR-100, and TinyImageNet, with all images downsampled to $32 \times 32$ pixels, deemed a "toy problem" by 2026 standards. It is highly uncertain whether robustness features learned on $32 \times 32$ images, and the CKA-based structural alignment that transfers them, are in any way representative of the features and alignment challenges in high-resolution, large-scale vision foundation models.
3. The language experiments use models that are orders of magnitude smaller than the popular models (e.g. SmolLM2-135M/360M and GPT2-Large v.s. Qwen3-8B). The paper itself admits these are "relatively small". The gains on these tiny models provide very little evidence of practical utility because scaling up to multi-billion parameter models introduces alignment challenges and it's a significant leap to assume this method would be effective.

**Questions:**

N/A

---

> ### Author Response · Authors · 2025-11-21
>
> Dear reviewer rkTA,
>
> We greatly appreciate your feedback.  Below, we respond to your concerns about layer selection, use of lower resolution vision datasets, and smaller scale language models.
>
> **Regarding layer selection**
>
> To avoid an exhaustive search while identifying target alignment layers, we used a lightweight probing procedure that evaluates a number of candidate layers (appendix section A.1.4) to select guiding and guided alignment layers in BIRD.  We found this simple approach reliably identified strong alignment layers and observed that BIRD’s performance is not overly sensitive to the exact layer choice (Appendix Figure 7).  These probing tools will be released in our public repository to help practitioners bootstrap their own layer selection process when applying BIRD to new contexts.
>
> Although multi-layer alignment is a natural extension, it is certainly nontrivial as it introduces many new hyperparameters including the selection of additional teacher-student distillation layer pairs and how the representation loss should be weighted at each layer, as behavior is typically encoded non-uniformily across a model [1,2].  We have made preliminary attempts at mutli-layer alignment by naively selecting four layers within the teacher and student, at regularly spaced depths throughout each network, and aligning their representation structures with equal weight distillation loss using BIRD.  In these experiments, we typically observed performance benefits similar to those obtained with single-layer BIRD, but not consistently better or worse.  Results and commentary around this have been added to Sections 8 and A.3 of the revised paper.
>
> [1] Zhou et al., 2025. “Representation Engineering: A Top-Down Approach to AI Transparency.”
>
> [2] Vig at al., 2020. “Investigating Gender Bias in Language Models Using Causal Mediation Analysis.”
>
>
> **Regarding the use of low resolution images**
>
> **Justification:** We chose to use lower-resolution images in our TinyImageNet and ImageNet experiments to enable a large-scale, systematic evaluation of BIRD.  Between the analyses of Sections 4 and 6, over 850 vision models were trained.  Using full-scale ImageNet images in this analysis unfortunately was not feasible with our available compute.
>
> We additionally believe this choice was justified as previous work has demonstrated that performance on lower resolution datasets can often serve as reliable, lower-cost proxies for performance scaling on higher resolution datasets [3, 4].
>
> [3] Zoph et al., 2018. “Learning Transferable Architectures for Scalable Image Recognition.”
>
> [4] Chrabaszcv et al., 2017. “A Downsampled Variant of ImageNet as an Alternative to the CIFAR datasets.”
>
> **Follow-up experiment using BIRD with high resolution images and foundation models:** In follow-up experiments, we demonstrate the efficacy of BIRD when used with foundation vision models and higher resolution imagery.   Specifically, we consider the application of distilling the semantically structured and generalizable representations from CLIP pre-trained models into a lightweight ConvNeXt ImageNet classifier (as CLIP pre-trained models have been shown to more generalizable and semantically structured representations than ImageNet-trained models [5]).  In short, we find that BIRD can indeed be utilized to help transfer such generalizable representations to the ConvNeXt student (as evaluated by held-out classification accuracy on the ImageNet-Sketch dataset).  Further details can be found in Section A.6 of the Appendix.
>
> We suggest that these results provide evidence for the utility of BIRD in the context of modern, vision foundation models and higher-resolution imagery.
>
> [5] Radford et al., 2021. “Learning Transferable Visual Models From Natural Language Supervision.”
>
>
> **Regarding the use of small language models**
>
> We do acknowledge that the evaluated language models are not on the same scale of multi-billion parameter models and seek to clearly acknowledge this in Section 7.  Unfortunately, evaluating BIRD on multi-billion parameter models is not possible with our available compute.
>
> That said, we find it crucially important not to overlook the efficacy of methods (BIRD and others) on “smaller” (relative to multi-billion parameter) models.  Despite the surge in popularity of multi-billion parameters models, smaller models (on the order of $10^7$-$10^8$ parameters) are ubiquitously used on consumer devices, desktops, mobile devices, IoT devices, robotics, etc.  We systematically demonstrate the efficacy of BIRD on models of this scale.  BIRD is especially well positioned for transferring aligned behaviors from the ever-expanding pool of large teacher models into smaller models without access to the teacher’s training data and costly curated datasets.  We see scaling BIRD to multi-billion parameter models as a natural extension, rather than a requirement for demonstrating its utility.

---

### Official Review · Reviewer_1C7Y · 2025-10-31

**Soundness:** 4
**Presentation:** 3
**Contribution:** 4
**Rating:** 6
**Confidence:** 5

**Summary:**

The paper introduces BIRD, a method to transfer behavioral properties like robustness by matching the representation structure (via CKA) between a teacher and a student model. The central claim is that this enables transfer across different tasks, datasets, and architectures. The authors provide comprehensive experiments, primarily in robust image classification, showing BIRD outperforms strong transfer learning baselines. A large-scale analysis identifies interpretable properties of teacher representations that predict transfer success. The paper is complemented with LLM experiments, showcasing the method's generalizability. The idea is novel and can indeed be a practical tool for achieving desirable representation structures within various models.

**Strengths:**

1. I believe the main idea is quite solid, using the geometry of the representation is justified, and the authors show that it also yields practical utility, beyond the intuitive justification. The overall hypothesis has high potential impact.
2. Identifying predictive factors is a valuable contribution. The high explained variance (up to 85%) provides practitioners with a principled way to select teachers.
3. The paper is well written and clearly explained, which helps with reproducibility.
4. The robust image classification experiments are extensive, covering many model architectures and dataset pairs. Hinting at solid results.

Overall, I believe the idea is quite solid and has high potential impact. I would like to see this paper accepted; However, I have a few concerns, which I will list below. I believe addressing these can indeed increase its potential.

**Weaknesses:**

There is a lack of ablation studies on some key aspects, or at least some important metrics are not reported.

Major:
1. The batch size B is a fundamental hyperparameter for CKA. This is not reported anywhere. The performance could be highly sensitive to this parameter.
Minor:
2. The choice of kernel (e.g., linear vs. RBF) can drastically change the geometry being compared. No ablation or discussion is provided on why this specific similarity measure was chosen over others.
3. The paper does not explore multi-layer distillation, which is a natural extension.
4. The values for $\alpha$ and $\beta$ should be reported.

Another problem would be with some claims, which are not justified by current experiments.

Major:
4. The paper repeatedly emphasizes that BIRD works "even when the teacher and student differ in architecture, task, and training data" and "does not require a shared input space." However, all experiments use the same input modality and resolution (images resized to 32x32). The teacher and student process the same input batches. The method has not been demonstrated to work across truly different input spaces (e.g., teacher on text, student on images; teacher on high-res images, student on low-res images). This is a significant overstatement of the current evidence.

**Questions:**

1. Could you provide more experimental details, especially the batch size used, and how it affects the performance?
2. Were other design choices, e.g., different kernels, multi-layer distillation, explored? If not, are there any reasons why we should not look into those, or is it just not that beneficial to do so?
3. What would happen if we use some pretrained teacher models? e.g., foundation encoders like CLIP would indeed be good examples of models with aligned representations, which could also be good for assessing the claim that "BIRD does not require a shared input space.".

---

> ### Author Response · Authors · 2025-11-21
>
> Dear reviewer 1C7y,
>
> Thank you for your review. We greatly appreciate these questions, feedback, and suggestions for improvements. We are encouraged by your positive assessment of our work’s contribution and soundness and seek to address your remaining questions and concerns below.
>
> **Batch size ablation**
>
> We have run an ablation study over the batch size used in BIRD and find that behavior is transferred much more effectively for larger batch sizes, as expected.  This supplemental analysis can be found in Section 5 of the revised paper.  In short, we find that while smaller batch sizes (e.g., 32, 64) offered performance improvements in robust behavior transfer over the original student model, those improvements were far less than could be achieved with larger batch sizes. The experimental results reported in our main text rely on a batch size of 128, though there is room for even further improvement when using larger batch sizes than this.
>
> **CKA Kernel choice**
>
> Our choice of a linear kernel in the CKA-based distillation was selected based on its prior success in evaluating similarities in deep learning representations [1,2] and its ease of interpretation.  That said, in supplemental experiments we have also observed successful behavior transfer when using an RBF kernel with a bandwidth, $\sigma$, comparable to that achieved with a linear kernel.  Experimental results have been added to Section 5 and Appendix Section A.1.7 of our revised paper.
>
> More broadly, a key contribution of BIRD is a flexible distillation framework grounded in representation structure.  In settings where specific geometric properties of the representation space are known to be behaviorally relevant a priori, we hope that practitioners may be inspired to adapt BIRD with such an application-specific kernel.
>
> [1] Kornblith et al., 2019. "Similarity of Neural Network Representations Revisited."
>
> [2] Nguyen et al., 2021. "Do Wide and Deep Networks Learn the Same Things? Uncovering how Neural Network Representations Vary with Width and Depth"
>
> **Selection of alpha and beta**
>
> Thank you for pointing this out that alpha and beta were not reported.  We have added these details to Appendix Sections A.1.3 and A.5.2 of the revised paper.
>
> **Multi-layer alignment**
>
> Two primary challenges exist with multi-layer alignment: (1) selecting distillation layer pairs across the teacher and student and (2) how the representation structure distillation loss should be weighted at each layer, as behavior is typically encoded non-uniformily across a model [3].  We have made preliminary attempts at mutli-layer alignment by naively selecting four layers within the teacher and student, at regularly spaced depths throughout each network, and aligning their representation structures with equal weight distillation loss using BIRD.  In these experiments, we typically observed performance benefits similar to those obtained with single-layer BIRD, but not consistently better or worse.  While multi-layer alignment is a natural extension, we find that it is certainly not trivial given the number of hyperparameters it introduces.  Results and commentary around this have been added to Sections 8 and A.3 of the revised paper.
>
> [3] Zhou et al., 2025. "Representation Engineering: A Top-Down Approach to AI Transparency"
>
>
> **Claims surrounding BIRD not requiring a shared input space and suggested experiments with CLIP**
>
> Thank you for pointing this out.  By claiming that the teacher and student do not require a shared input space, we intended to suggest that the teacher and student did not require a shared input *domain* (e.g., the teacher is trained on a narrow classification domain, like CIFAR-10, whereas the student's target task is classification on a much broader input domain of natural images, like ImageNet).  We acknowledge this discrepancy and have updated this claim throughout the paper, replacing input "space" with "domain".  We appreciate you highlighting this.
>
> We were very excited about your idea of learning from CLIP foundation encoders and have prepared experiments to investigate this.  We consider the application of distilling the semantically structured and generalizable representations from CLIP pre-trained models into a lightweight ConvNeXt ImageNet classifier.  In short, we find that BIRD can indeed be utilized to help transfer such generalizable representations to the ConvNeXt student (as evaluated by held-out classification accuracy on the out-of-distribution, ImageNet-Sketch dataset).  Further details of this experiment and results can be found in Section A.6 of the Appendix (and referenced in discussion commentary).
>
> We greatly appreciate your suggestion for running this type of experiment.  Not only do we think it further highlights the generality of BIRD, providing initial evidence of the teacher and students not requiring a shared input *space*, but are also excited by the application of BIRD in the context of modern vision foundation models.

---

### Author Response · Authors · 2025-12-03
**Summary for Area Chairs [1/2]**

Dear Area Chairs,

We greatly appreciate your time and effort throughout this review process. In light of the updated review process, we have provided a concise summary of the feedback and commentary from our four reviewers, below, along with our responses. We hope that this summary is helpful to understand the strengths, questions, and concerns raised by reviewers, as well as the clarifications and new analyses that we have added during the rebuttal period to our revised manuscript.

First, we are very grateful and motivated by the support from our reviewers.  Notably, we are pleased by several of the strengths that the reviewers have highlighted:

- **Core idea and high potential impact:** Reviewers provided positive feedback for our central hypothesis of transferring aligned behavior between non-homogenous model pairs via representation-structure distillation (1C7Y, W2qD), the broad applicability of this hypothesis (rkTA, X3w8), and its novelty (1C7Y, X3w8).  Review 1C7Y explicitly acknowledges our method as novel, well-justified, intuitive, and having high potential impact, mentioning *“Overall, I believe the idea is quite solid and has high potential impact. I would like to see this paper accepted;”*.
- **Strong empirical foundations:**  Reviewers acknowledge that the results of our extensive experimentation support our central claims for the utility of BIRD, mentioning *“The robust image classification experiments are extensive, covering many model architectures and dataset pairs. Hinting at solid results.”* (1C7Y) and that our experiments *“Demonstrates versatility beyond vision”* (X3w8) and *“demonstrates its potential as a general tool”* (rkTA).
- **Broad applicability and flexibility:** Reviewers agreed that BIRD is flexible and scalable, and noted its promising applications to vision, safety alignment, and weak-to-strong generalization.  rkTA specifically acknowledges that *“The paper successfully shows BIRD is not just a vision technique. Its application to DPO (safety) and soft-label distillation demonstrates its potential as a general tool.”* and *“BIRD does not require the teacher and student to share an input space, output space, task, or architecture”*.  Meanwhile, reviewer X3w8 suggests *“Unlike traditional knowledge distillation or continual learning, BIRD doesn't require shared tasks, data, or output spaces, allowing transfer from small/simple teachers”*.
- **Practical utility and value of teacher-selection analysis:** Reviewers noted the practical value and important contribution of our large-scale analysis that uncovered interpretable properties of teacher representations that enable successful behavior transfer.  Reviewer X3w8 specifically states that this analysis makes *“the method actionable and predictable, advancing beyond ad-hoc approaches.”* and reviewer 1C7Y suggests *“Identifying predictive factors is a valuable contribution”* and that this analysis *“provides practitioners with a principled way to select teachers”*.
- **Clarity and reproducibility:** Several reviewers provide high ratings for presentation clarity.  Reviewer 1C7Y specifically mentions *“The paper is well written and clearly explained, which helps with reproducibility.”*

---

> ### Author Response · Authors · 2025-12-03
> **Summary for Area Chairs [2/2]**
>
> We are also greatly appreciative of the questions and concerns that were raised. Follow-up experiments and revisions that were prompted by this feedback played a valuable role in improving our manuscript. We provide a brief summary of key points from this commentary, follow-up experiments, and associated manuscript revisions that diligently address each of these points below:
>
> - **Ablation studies:** Reviewers 1C7Y and W2qD asked about the role of batch size in BIRD. We conducted a dedicated, follow-up experiment to answer this question and found that while small batches still yield improvements over the baseline student model, robust behavior transfer strengthens with larger batches, as expected.
> Reviewer 1C7Y also asked about CKA kernel choice. In follow-up experiments over 12 settings, we observed that Linear and RBF kernels provided comparable robustness gains. Additional results and discussions from these new studies were added to Section 5 and A.1.7 of the revised manuscript.
> - **Layer selection and multi-layer alignment:** Reviewers rkTA and X3w8 asked about potential challenges of BIRD layer selection. Importantly, we clarify that our layer selection strategy was intentionally designed to avoid exhaustive search and to be easily adapted to new contexts. We show BIRD is not overly sensitive to the exact layer choice and provide further details on layer selection in A.1.4.
> Reviewers rkTA and 1C7Y also commented that multi-layer distillation was an unexplored extension in the original submission. We agree this is an interesting direction for future work, but show in follow-up experiments that this is a non-trivial extension due to the number of hyperparameters it introduces (Sections 8 and A.3 of the revised manuscript).
> - **Claims surrounding BIRD not requiring shared input spaces and new experiments with CLIP encoder:** Reviewer 1C7Y identified an imprecise wording in our original submission, where we claim that BIRD enables behavior transfer between models that do not share the same input *space* but actually intended to suggest that the models do not require a shared input *domain* (directly supported by our results).
> We have resolved this concern by (1) updating the wording in our revised manuscript accordingly and (2) running a new follow-up experiment, where we demonstrate effective behavior transfer from a vision-language CLIP encoder using BIRD, following the suggestion of 1C7Y. The results of this experiment (detailed in A.6) successfully demonstrate that the teacher and student do not require a shared input space, further demonstrating the generality of BIRD.
> - **Use of low resolution images:** Reviewer rkTA suggested our robustness experiments were limited due to the use of low-resolution images.  We chose to use lower-resolution images in these experiments to enable large-scale, systematic evaluation of BIRD. Previous work has demonstrated that performance on lower resolution datasets can serve as reliable, lower-cost proxies for performance scaling on high resolution datasets [^1,^2].  Finally, our newest experiments transferring behavior from CLIP encoders provide direct evidence for the utility of BIRD with high-resolution images.
> - **Computational overhead:** X3w8 asked about the computational overhead of BIRD.  We have provided computational complexity analyses and a new table with quantitative evidence that the CKA computation in BIRD is relatively inexpensive and requires *fewer* floating point operations in the evaluated contexts as compared to logits-based knowledge distillation.
> - **Transferred property analysis:** Reviewer W2qD asked about properties transferred from teacher to student in the robust behavior transfer experiments.  We performed additional analyses and show that improved robustness in the BIRD student emerges from learning features are more stable in the presence of out-of-distribution perturbations.
> - **Novelty:** Reviewer W2qD claimed limited novelty and application scope of BIRD as compared to learning directly from teacher outputs and suggested that the main selling point of BIRD is that it is agnostic to representation dimensionality. These claims miss many of the core and novel contributions of BIRD: BIRD establishes a mechanism for aligned behavior transfer that operates across domains, label spaces, and architectures by supervising over representation structure alone (in stark contrast to existing approaches in knowledge distillation and alignment, which assume a shared output spaces between teacher and student, access to labeled data from the teacher’s domain, and/or availability of costly, curated datasets to define an alignment objective with task-level supervision).  These novelties were explicitly noted by X3w8 and 1C7Y.
>
> [^1] Zoph et al., 2018. “Learning Transferable Architectures for Scalable Image Recognition.”
>
> [^2] Chrabaszcv et al., 2017. “A Downsampled Variant of ImageNet as an Alternative to the CIFAR datasets.”

---

### Meta-Review · Area_Chair_1GzU · 2026-01-03

**Summary:**

The paper proposes BIRD, a method for behavior transfer using a knowledge distillation approach, where Centered Kernel Alignment (CKA) is used to measure distances between student and teacher representations. Specifically, the student (typically stronger than teacher) is finetuned using a mix of its task loss (e.g. a normal classification loss on ImageNet1k) and a distance between teacher and student representations measured via CKA. The authors perform broad evaluations covering robustness transfer in vision (robust teacher on CIFAR-10 -> student on ImageNet-1k with no robustness training), alignment transfer in small language models and weak-to-strong generalization (text classification).

First, I would like to note that feature-based knowledge distillation is a very active area of research, and CKA is a common tool in this literature.  See for example [1] which explicitly uses CKA for distillation. Works [2-4] are some other recent examples of feature-based distillation, none of which are discussed in the paper. The authors state:

> available KD methods have since expanded to include intermediate-layer supervision (Romero et al., 2014; Zagoruyko & Komodakis, 2016) and cross-modal transfer (Gupta et al., 2016).
These methods assume shared tasks and output spaces and often transfer sample-level information.

I believe the main difference between the paper under consideration and the existing feature-based distillation work is in the evaluation setting: the paper considers cross-task transfer of properties. I am not immediately aware of other papers using feature distillation for this goal, although I would not be surprised if papers like that existed.

Another point I would like to raise is that the original weak-to-strong generalization paper considers a weak model as an analogy for a human aligning a superhuman AI system; in that setting, the method proposed by the authors is not applicable: we cannot compute a CKA between a human and an AI model. So interventions based on feature alignment are generally not applicable to the setting considered in the original weak-to-strong paper, although that does not disqualify the results of the paper, and it may have other applications.

Reviewer W2qD pointed to the limited novelty of this work. Reviewers rkTA and W2qD argue that the experiments in the paper are overly toy and are not relevant to practical robustness training. The experiments on language models are also relatively toy. As a result, it is unclear if the method will be impactful practically.

**References**

[1] _Rethinking Centered Kernel Alignment in Knowledge Distillation_. Zikai Zhou, Yunhang Shen, Shitong Shao, Linrui Gong, Shaohui Lin ; IJCAI-24. [pdf](https://www.ijcai.org/proceedings/2024/0628.pdf)

[2] _Better Teacher Better Student: Dynamic Prior Knowledge for Knowledge Distillation_; Zengyu Qiu, Xinzhu Ma, Kunlin Yang, Chunya Liu, Jun Hou, Shuai Yi, Wanli Ouyang.

[3] _Exploring Inter-Channel Correlation for Diversity-preserved Knowledge Distillation_;
Li Liu, Qingle Huang, Sihao Lin, Hongwei Xie, Bing Wang, Xiaojun Chang, Xiaodan Liang

[4] _A Comprehensive Overhaul of Feature Distillation_;
Byeongho Heo, Jeesoo Kim, Sangdoo Yun, Hyojin Park, Nojun Kwak, Jin Young Choi

**Reviewer Concerns:**

Several concerns of the reviewers have been addressed in the rebuttal:
- Additional ablations on the hyper-parameters of the method
- Using representations across multiple layers and layer selection
- Using images of different resolutions
- Transfer from a general-purpose CLIP model
- Concerns about computational overhead of the method
- Mechanistic analysis on the transfered properties

The following concerns of the reviewers are still standing:
- Novelty
 + I agree with this concern, I believe the literature on feature-based knowledge distillation is not adequately discussed in the paper currently
- Evaluation quality, experiments too toy
 + I agree with this concern, adding that the weak-to-strong evaluation in Section 7.2 is not very meaningful at least in the original weak-to-strong generalization setting.

The paper has some additional merits, e.g. there is an analysis on the impact of the teacher on the results of the transfer in Section 6. I would also like to highlight that while the experiments are relatively toy, they are quite extensive, covering many models and datasets.

Generally, I think the paper is borderline: the novelty is limited and the experiments are not necessarily sufficient evidence that the method will be useful in practice. However, the paper presents some interesting results and analysis, that could be of interest to the ICLR community.

**Reviewer Scores:**

- W2qD: 4 → 4 (concerns on novelty and limited evaluation not fully addressed)
- X3w8: 6 → 6, possibly 8 (did not have major concerns, questions answered in rebuttal)
- rkTA: 4 → 4 (concerns on the limited evaluation not fully addressed)
- 1C7Y: 6 → 6, possibly 8 (did not have major concerns, questions answered in rebuttal)

---

### Decision · Program_Chairs · 2026-01-26

Accept (Poster)